# Kistamicin biosynthesis reveals the biosynthetic requirements for production of highly crosslinked glycopeptide antibiotics

Anja Greule [1,2], Thierry Izoré[1,2,9], Dumitrita Iftime[3,9], Julien Tailhades[1,2,9], Melanie Schoppet[1,2], Yongwei Zhao[1,2], Madeleine Peschke[4], Iftekhar Ahmed [5], Andreas Kulik [3], Martina Adamek[3], Robert J.A. Goode [1,6], Ralf B. Schittenhelm[1,6], Joe A. Kaczmarski[7], Colin J. Jackson [7], Nadine Ziemert[3], Elizabeth H. Krenske[5], James J. De Voss [5], Evi Stegmann[3,8] & Max J. Cryle [1,2]

Kistamicin is a divergent member of the glycopeptide antibiotics, a structurally complex class of important, clinically relevant antibiotics often used as the last resort against resistant bacteria. The extensively crosslinked structure of these antibiotics that is essential for their activity makes their chemical synthesis highly challenging and limits their production to bacterial fermentation. Kistamicin contains three crosslinks, including an unusual 15-membered A-O-B ring, despite the presence of only two Cytochrome P450 Oxy enzymes thought to catalyse formation of such crosslinks within the biosynthetic gene cluster. In this study, we characterise the kistamicin cyclisation pathway, showing that the two Oxy enzymes are responsible for these crosslinks within kistamicin and that they function through inter-actions with the X-domain, unique to glycopeptide antibiotic biosynthesis. We also show that the kistamicin OxyC enzyme is a promiscuous biocatalyst, able to install multiple crosslinks into peptides containing phenolic amino acids.

[1] Department of Biochemistry and Molecular Biology, The Monash Biomedicine Discovery Institute, Monash University, Clayton, VIC 3800, Australia. [2] EMBL Australia, Monash University, Clayton, VIC 3800, Australia. [3] Interfaculty Institute of Microbiology and Infection Medicine Tübingen, Microbiology/Biotechnology, University of Tübingen, Auf der Morgenstelle 28, 72076 Tübingen, Germany. [4] Department of Biomolecular Mechanisms, Max Planck Institute for Medical Research, Jahnstrasse 29, 69120 Heidelberg, Germany. [5] Department of Chemistry, The University of Queensland, St Lucia, QLD 4072, Australia. [6] Monash Biomedical Proteomics Facility, Monash University, Clayton, VIC 3800, Australia. [7] Research School of Chemistry, The Australian National University, Acton, ACT 2601, Australia. [8] German Centre for Infection Research (DZIF), Partner Site Tübingen, 72076 Tübingen, Germany. [9] These authors contributed equally: Thierry Izoré, Dumitrita Iftime, Julien Tailhades. Correspondence and requests for materials should be addressed to E.S. (email: evi.stegmann@biotech.uni-tuebingen.de) or to M.J.C. (email: max.cryle@monash.edu)

The glycopeptide antibiotics (GPAs) are a series of complex, peptide-based antibiotics that have been utilised in a clinical setting since the late 1950s and are exemplified by vancomycin[1]. Structurally, GPAs consist of a heptapeptide backbone that is heavily crosslinked via the side chains of aromatic amino acids within the peptide sequence (Fig. 1a)[1]. The crosslinking of the peptide backbone rigidifies the GPA peptide aglycone, allowing this to form a non-covalent complex with the dipeptide terminus of the cell wall precursor lipid II. By binding lipid II, GPAs are able to prevent the formation of the bacterial cell wall, hence leading to their antimicrobial activity[1]. The crosslinks within GPAs are therefore essential for the activity of these antibiotics and in addition are also the most challenging part of the total synthesis of these important molecules[2]. Given the difficulties of their commercial synthesis through traditional synthetic means, all current production of GPAs remains reliant on the bacterial fermentation of the natural producer strain, which in turn makes understanding this machinery of great practical as well as scientific interest[3].

GPAs are produced by the actions and interplay of two powerful biosynthetic systems. First, a non-ribosomal peptide synthetase (NRPS) machinery produces the linear heptapeptide GPA backbone, the core of which largely consists of non-proteinogenic phenylglycine residues[4–6]. Second, an oxidative cyclisation cascade consisting typically of three to four cytochrome P450 (Oxy) enzymes (one enzyme per ring formed, see Fig. 1b) is then recruited to the NRPS-bound heptapeptide in order to perform the stepwise cyclisation of the peptide into a rigid, active antibiotic[7,8]. The recruitment of the Oxy enzymes to the NRPS machinery is mediated by a unique P450 recruitment domain within the NRPS, known as the X-domain[9]. Different GPAs are classified based on the residues found in their peptide backbone, the crosslinking pattern of the final GPA and the modifications installed following construction of the crosslinked peptide aglycone[1]. Most common GPAs with appreciable antibiotic activity are found in Types I-IV: Type I/II include GPAs with three crosslinks within the peptide backbone and aliphatic or aromatic residues at positions 1/3, respectively. Type III and IV GPAs contain an additional crosslink between aromatic residues at positions 1/3 of the peptide, with Type III/IV differentiated by the absence or presence of an aliphatic chain linked to a peripheral sugar moiety (Fig. 1a)[1].

Due to the clinical utility of GPAs and the challenges associated with the synthesis of these complex molecules, significant research efforts have been made to understand the cyclisation process of GPAs from both an in vivo[3,10,11] and in vitro standpoint[9,12–15]. These studies have revealed that each Oxy enzyme is responsible for the installation of one crosslink at the heptapeptide stage (Fig. 1b) and that a specific order of activity exists (OxyB, (±OxyE), OxyA, OxyC). Given the importance of the GPA cyclisation cascade, identifying new members of related biosynthetic machineries is key to understanding the biocatalytic potential of these powerful catalysts: for this reason, there is great interest in an unusual class of GPAs—known as Type V GPAs—that include the representative members complestatin[16,17] and kistamicin[18,19]. Type V GPAs, whilst the most structurally divergent GPA class, share sufficient similarities in their structure and biosynthesis to be classed as GPAs in spite of their lack of glycosylation and reduced antibacterial activity (MIC values reported for complestatin and kistamicin against *S. aureus* of ~2 μg.mL$^{-1}$/12.5–25 μg.mL$^{-1}$ as opposed to 0.5–1 μg.mL$^{-1}$ for vancomycin)[19–22]. Type V GPAs display divergent activity when compared to Type I–IV GPAs, including antiviral activity for both kistamicin[19] and complestatin[20,21] as well as other types of potential antibacterial activity[22]; this clearly makes such atypical GPAs of great interest for diversification of the activity of GPAs.

One major structural difference in Type V GPAs is the DE aryl crosslink formed from the tryptophan residue at position 2 of the heptapeptide, whilst kistamicin in particular also contains an additional unusual ring, an A-O-B ring formed from a C-terminal 4-hydroxyphenylglycine (Hpg) residue[1,5]. Furthermore, the kistamicin gene cluster only contains two Oxy enzymes despite the presence of three crosslinks in the final product, whilst the NRPS machinery also contains features that are not found in other GPAs. Given these differences in structure and biosynthetic machinery found in the oxidative crosslinking cascade of kistamicin biosynthesis, we have engaged in a detailed structural, biochemical and functional characterisation of this crosslinking cascade in this unique GPA.

Here, we show that the three crosslinks that occur during the kistamicin peptide construction are catalysed by only two Oxy enzymes, with the activity of the kistamicin Oxy enzymes being distinct from those found in the biosynthesis of typical Type I-IV GPAs. Despite these differences, the kistamicin Oxy enzymes still rely on recruitment to the peptide bound to the NRPS by interaction with the X-domain, a unique recruitment domain found in GPA biosynthesis. Through the biochemical studies and structural characterisation of an OxyA$_{kis}$/X-domain complex we can now show that not only is the X-domain mediated Oxy recruitment mechanism conserved across all known types of GPA biosynthetic machineries but also that the interface between all Oxys and the X-domain is conserved, supporting a shuffling mechanism for Oxy activity. Within the kistamicin cyclisation cascade, we show that the kistamicin OxyC enzyme (OxyC$_{kis}$) acts first, catalysing insertion of the C-O-D ring, which is followed by the actions of OxyA$_{kis}$ installing the DE aryl crosslink. Whilst it is not possible to unambiguously assign the insertion of the A-O-B ring to OxyC$_{kis}$ activity, supporting evidence from our in vivo and in vitro activity studies indicates that OxyC$_{kis}$ is a highly promiscuous enzyme that is capable of peptide bicyclisation and hence the most probable candidate for this biotransformation.

## Results

**The *kis* cluster reveals a divergent biosynthetic apparatus.** Due to the unusual structure of kistamicin when compared with typical type I–IV GPAs, we initially sequenced the genome of *Actinomadura parvosata* subsp. *kistnae* (*Nonomuraea* sp. ATCC55076) to identify the kistamicin biosynthetic gene cluster. The kistamicin (*kis*) biosynthetic gene cluster spans ~60 kb with 33 open reading frames (Supplementary Fig. 1). During the course of our investigations, this cluster was also revealed by the Seyedsayamdost[23] and Gulder groups[24]. Strikingly, the biosynthetic gene cluster of this GPA encodes only two cytochrome P450 enzymes (KisN and KisO) implicated in the crosslinking of the aromatic side chains of the peptide, despite there being three crosslinks in kistamicin. A phylogenetic analysis of the Oxy enzymes from kistamicin biosynthesis revealed that these fall into the OxyA (KisN) and OxyC (KisO) families and do not cluster with OxyB, the first enzyme involved in cyclisation cascade of the Type I-IV GPAs (Supplementary Fig. 2). This was unexpected, given that OxyB-catalysed insertion of the C-O-D ring is an essential prerequisite for the activity of subsequent Oxy enzymes in Type I–IV systems[9–11,25]. The OxyA$_{kis}$ enzyme is expected to install the DE ring based on comparison to the complestatin system[26,27], which led us to hypothesise that the OxyC$_{kis}$ enzyme was responsible for installation of both the A-O-B as well as C-O-D rings in kistamicin.

Beyond the Oxy enzymes, the kistamicin NRPS machinery maintains the unique GPA X-domain for Oxy recruitment to the NRPS-bound peptide[1,3], but contains several differences to typical GPA NRPS assembly lines, including in modules 1

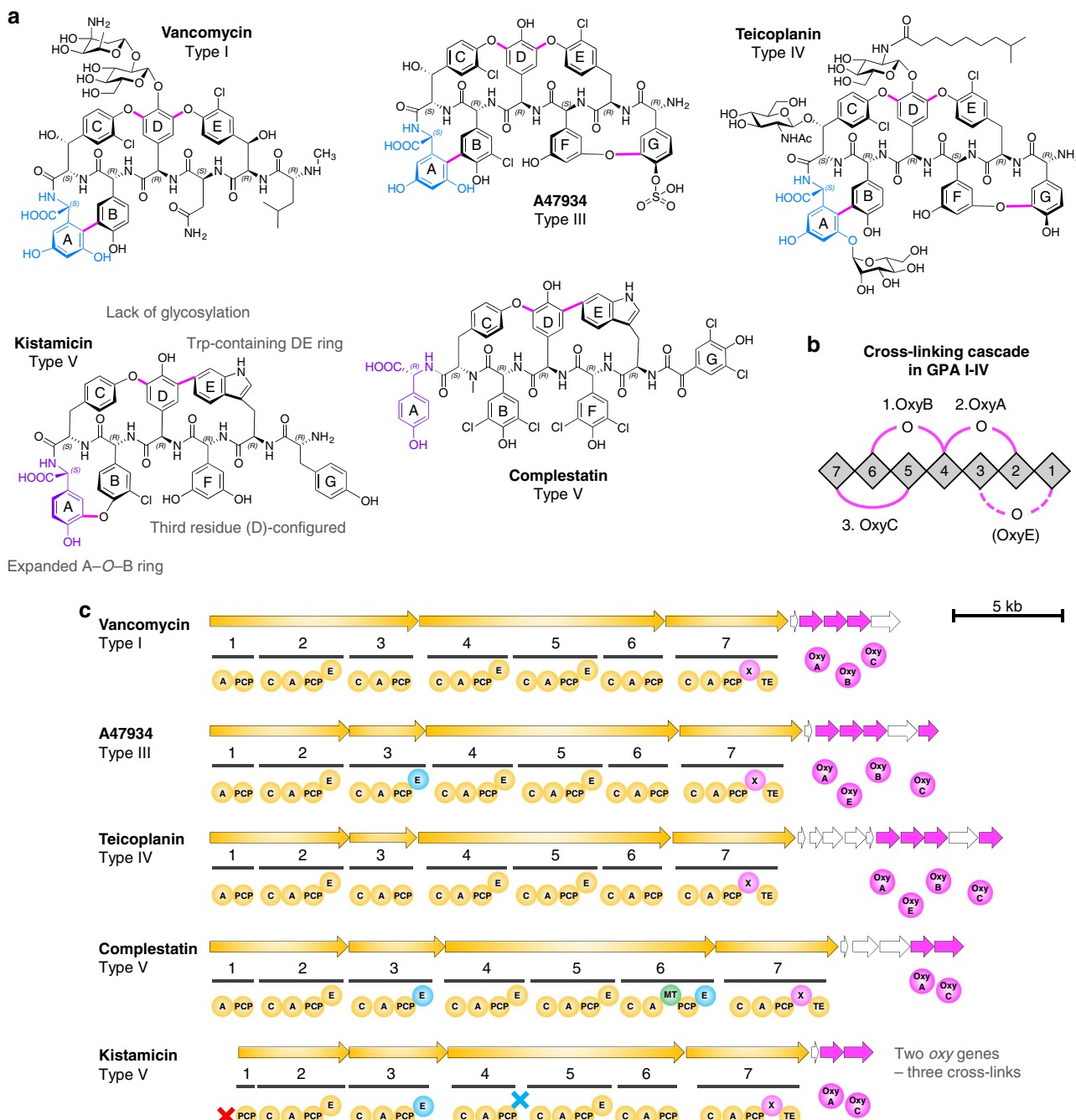

**Fig. 1** GPA structures and NRPS/Oxy interplay. **a** Structure of different types of GPAs shown for vancomycin (Type I), A47934 (Type III), teicoplanin (Type IV), kistamicin (Type V) and complestatin (Type V). Major structural differences between kistamicin and Type I–IV GPAs include: the replacement of the AB ring required for the activity of standard GPAs with an enlarged A-O-B ring; the replacement of the D-O-E ring with a DE ring that incorporates a tryptophan residue; the opposite configuration of the 3rd peptide residue (D instead of L); and the lack of glycosylation. C-terminal residues of GPAs are coloured for 3,5-dihydroxyphenylglycine (3,5-Dpg, light blue) and 4-hydroxyphenylglycine (4-Hpg, violet). P450 catalysed intramolecular crosslinks are indicated in pink. **b** P450-catalysed crosslinking cascade performed by first OxyB, (then optional OxyE), OxyA and finally OxyC into type I–IV GPA heptapeptides. Numbering of amino acids is listed according to the timing of amino acid incorporation, i.e. from 1–7. **c** Part of the respective GPA biosynthetic gene clusters involving the NRPS (yellow) and the P450 enzymes (pink); other genes are also indicated (white). The NRPSs are divided into seven modules with a number of catalytically active domains (yellow circles). In the last module, the Oxy-recruiting domain X is present (light pink). Additional domains (E domains (blue), MT domain (green)) and missing domains (red cross) are indicated. Presence and type of Oxys within the GPA biosynthesis is shown (pink circles). Kistamicin has three crosslinks, but only two Oxy encoding genes in the cluster, an OxyA and an OxyC enzyme. Domain definitions: A, adenylation domain; C, condensation domain; E, epimerisation domain; PCP, peptidyl carrier protein; MT, methyltransferase; X, Oxy-recruiting domain; TE, thioesterase domain

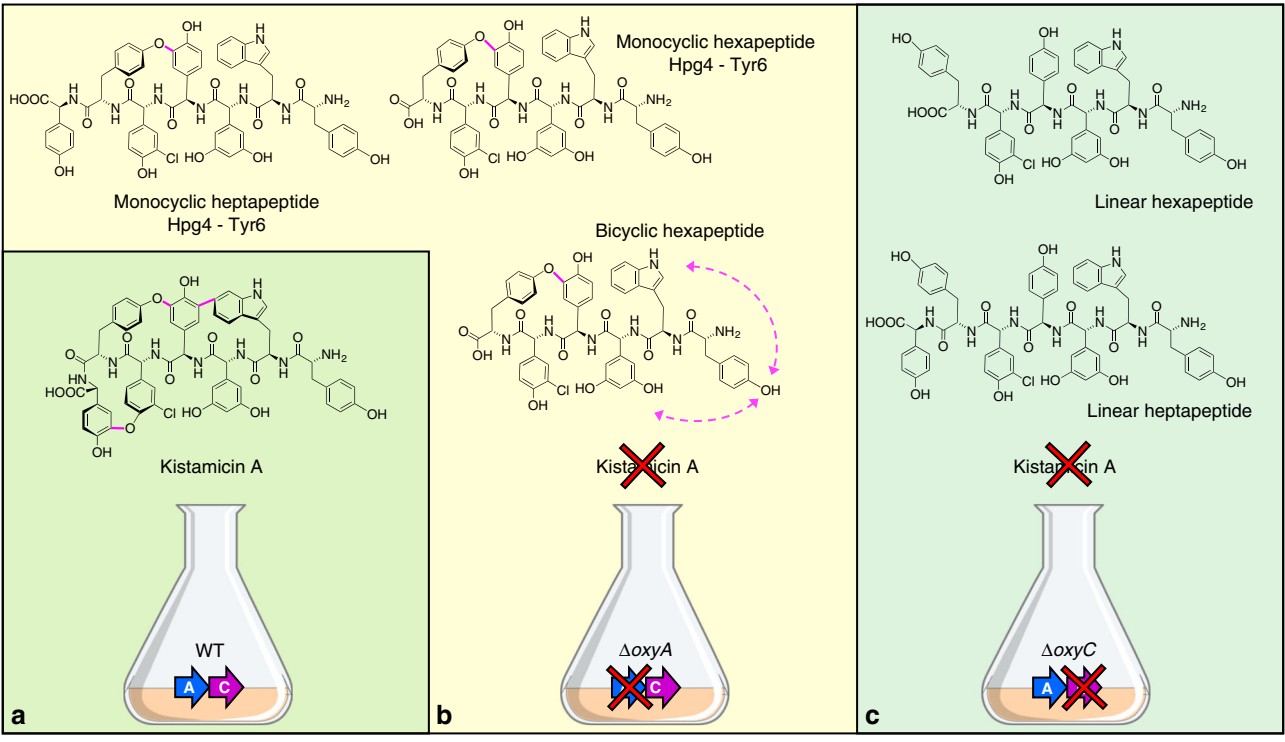

**Fig. 2** Production of kistamicin intermediates in *A. parvosata* WT and Δ*oxy* mutants. **a** *A. parvosata* wildtype (WT) produces kistamicin A, a tricyclic heptapeptide. **b** *A. parvosata* Δ*oxyA* (Δ*kisN*) produces several intermediates from kistamicin biosynthesis, including monocyclic hexa-/heptapeptides and different bicyclic hexapeptides (all structures contain the Tyr-6/ Hpg-4 C-*O*-D crosslink but display alternate off pathway cyclisation at the peptide N-terminus). **c** *A. parvosata* Δ*oxyC* (Δ*kisO*) produces only the linear hexa-/heptapeptide intermediates from kistamicin biosynthesis

(missing the core sub-domain of the adenylation domain), 3 (addition of an epimerisation (E)-domain) and 4 (missing an E-domain). The presence of an E-domain in module 3 is in agreement with the reported stereochemistry of the 3rd (Dpg) residue in kistamicin, whilst the lack of an E-domain in module 4 is highly unusual given the presence of a (D)-configured amino acid in position four of all GPAs characterized to date. Given the importance of this residue, alteration of the stereochemistry of this position to the (L)-form would appear highly unlikely (although would be necessary to explore, vide infra), thus implying a dual C/E function for one of the C-domains between modules 3/4 or 4/5. A phylogenetic analysis of the kistamicin C-domains did not reveal obvious clues to the identity of the dual C/E-domain, and confirmation of the identity of this domain awaits future characterisation (Supplementary Fig. 3).

**oxy gene deletions reveal the kistamicin cyclisation cascade.** Given the biosynthetic differences implicit in the sequence of the kistamicin gene cluster when compared to other GPA clusters, we first isolated and characterised kistamicin from the *A. parvosata* producer strain. Our structural analysis indicated that there was no difference between the structure of the compound that we had isolated and that previously reported (Supplementary Figs. 4–11, Supplementary Table 1)[18].

To ascertain the function of the Oxy enzymes within the kistamicin cluster, gene deletion mutants of OxyA_kis (*A. parvosata* Δ*kisN*) and OxyC_kis (*A. parvosata* Δ*kisO*) were prepared (Supplementary Figs. 12–16, Supplementary Table 2–4). After cultivation, kistamicin production was found to be abolished in both strains (Fig. 2, Supplementary Fig. 17). HRMS analysis of the isolates showed that the OxyC_kis deletion strain only produces linear hexa- and heptapeptides, whilst the OxyA_kis

deletion strain produced monocyclic hexa- and heptapeptides (Fig. 2). MS/MS analysis of these metabolites showed that the ring was inserted between Tyr-6 and Hpg-4, which corresponds to the C-*O*-D ring that is the first ring installed in all GPAs studied to date (Supplementary Figs. 18–23)[10,25,26]. This also indicates that the order of ring insertion activity matches that from other GPAs, with the DE ring is required before insertion of the A-*O*-B ring can occur[10,25,26]. The appearance of monocyclic hexapeptide is in agreement with experiments from other GPA producer strains that have shown that the initial ring insertion is the most facile and can occur at the hexapeptide stage if the NRPS machinery has stalled due to deletion of Oxy enzymes[28]. Production of kistamicin by the OxyA_kis (*A. parvosata* Δ*kisN*) deletion mutant strain was recovered through complementation of the gene on an integrative plasmid (Supplementary Figs. 17 and 19). These results support OxyC_kis acting first during the cyclisation of kistamicin, installing the C-*O*-D ring into the kistamicin peptide.

We next investigated lower abundance species in the OxyA_kis (*A. parvosata* Δ*kisN*) deletion mutant for evidence of bicyclic peptide formation. HRMS showed that low levels of bicyclic hexapeptide intermediates were indeed present in the OxyA_kis deletion strain (Fig. 2). Given that the seventh residue (involved in the A-*O*-B ring) is missing from this peptide, we next used MS/MS analysis to determine that a crosslink was formed between Tyr-1 and other N-terminal residues (Trp-2 and Dpg-3) in addition to the Tyr-6/ Hpg-4 C-*O*-D crosslink present in the monocyclic species (Supplementary Figs. 24–26). This suggests that the OxyC_kis enzyme can be promiscuous and is capable of installing two crosslinks in a single peptide, with the second ring then formed after initial C-*O*-D ring insertion. However, OxyC_kis is not able to form the DE ring, which in turn is required for A-*O*-B ring formation.

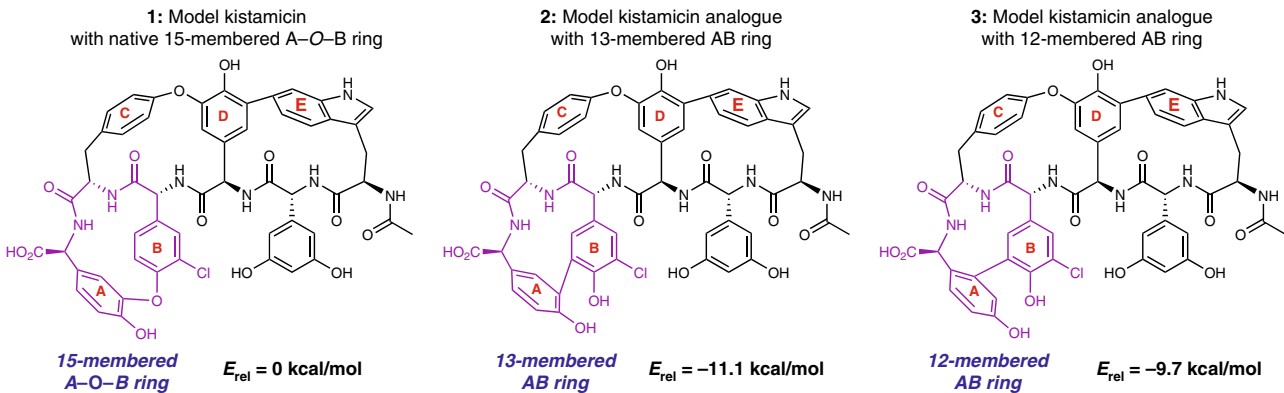

**Fig. 3** Computational analysis of alternative A-B ring crosslinks. Comparison of the relative energies of kistamicin (containing the native 15-membered A-O-B ring and modelled as **1**) relative to the energies of kistamicin analogues **2** and **3** which contain 13- and 12-membered AB rings, respectively. The ring derived from A-O-B crosslinking is significantly less stable than the smaller rings derived from AB crosslinking. Source data are provided as a Source Data file

Complementation of the previously published *oxyA* and *oxyC* deletion mutants from Type I (balhimycin)[25] and Type IV (A47934)[10] GPA producer strains by the kistamicin *oxyA_{kis}/ oxyC_{kis}* genes did not lead to recovery of antibiotic production in these strains as assessed by inhibition zone assays.

**Computational analysis of kistamicin crosslinking.** Based on these complementation studies, we next examined whether the inability of OxyC_{kis} to complement the Type I/IV OxyC knockout strains was due to differences in ring strain in the different GPA structures or whether the type of crosslink installed (aryl vs phenolic) was an inherent property of different Oxy enzymes. To understand whether the outcome of crosslinking, namely AB vs A-O-B ring formation is related in any way to the strain energies of the rings, we performed molecular mechanics and density functional theory computations (Fig. 3, see Supplementary Figs. 27–31 and Supplementary Information for details). Upon comparison of models of native kistamicin, featuring a 15-membered A-O-B ring (**1**), with two unnatural analogues containing 13- or 12-membered AB rings (**2/3**), we found unexpectedly that the A-O-B ring was significantly *less* stable (by about 10 kcal/mol) than the smaller-ring AB isomers. Interestingly, a similar observation was made when comparing the native 16-membered C-O-D ring of kistamicin with a 14-membered CD analogue. In this case, the 14-membered biaryl-linked CD structure corresponds to the known crosslinking pattern in the arylomycins. A third example mirroring the preference for biaryl linkages was observed when comparing the native 12-membered AB ring of Type I/IV GPAs with a 14-membered A-O-B analogue.

These results suggest that a low stability of phenolic crosslinks relative to diaryl crosslinks is a common feature among GPAs possessing Dpg or Hpg residues at the C-terminus. Indeed, this trend parallels the intrinsic energy difference between diphenyl ethers and biaryl alcohols (the latter being more stable than the former). In the case of kistamicin, an additional contributing factor to the high energy of the A-O-B ring is found to arise from ring strain; in particular, formation of this ring requires significant deformation of the chlorophenyl ether moiety. As such, these data indicate that the A-O-B ring in kistamicin is not a simpler ring to install than the AB ring present in standard GPAs. Rather, they support the hypothesis that the nature of the ring linkage installed is consistent for each individual Oxy enzyme, which is consistent with all in vitro and in vivo studies performed on GPA Oxy enzymes to date[10,25,26].

**Interaction of the Oxy enzymes with the kistamicin X-domain.** To study the potential interactions between the domains of the final NRPS module and the Oxy enzymes, we designed, expressed and purified OxyA_{kis}, OxyC_{kis}, the X_{kis}-domain alone as well as naturally fused to the preceding peptidyl carrier protein (PCP)-domain (PCP-X_{kis})[9,29]. Due to the similar size of the X-domain and Oxy enzymes, we performed N-terminal labelling of the X-domain with fluorescein to allow a specific wavelength to be monitored for each protein. These assays showed evidence of an interaction between the kistamicin Oxy enzymes and the X-domain independent of the presence of the adjacent PCP domain. This was evidenced by the formation of bands of higher molecular weight in the lanes where X-domain and Oxy enzymes were present (Fig. 4a–c). Whilst the common interaction partner is the X-domain itself, the native PAGE experiments also suggest that the presence of the PCP domain (albeit unloaded) may increase affinity for the Oxy enzymes.

With the limited resolution of these measurements, we next performed isothermal titration calorimetry (ITC) experiments to quantify the binding interactions occurring in the kistamicin system (Fig. 4d–f). These experiments showed that both Oxy enzymes associated with the X-domain, with the $K_d$ obtained for both in the low micromolar range (OxyC_{kis} 6.1 μM, OxyA_{kis} 10.4 μM). These values match the affinities determined for the teicoplanin system using alternative biochemical methods[9,30], and indicate that binding mostly driven by entropic effects, presumably due to the release of ordered water molecules from the interface. Given that the Oxy/NRPS interactions appear to be conserved despite the unusual structure of kistamicin and the non-standard crosslinking cascade, we next sought to understand this process at a molecular level.

**Characterisation of the kistamicin OxyA/ X-domain complex.** In order to gain further insight into the nature and sites of the Oxy/X-domain interactions from kistamicin biosynthesis, we were able to obtain crystals of a complex of the OxyA_{kis} protein and X-domain from kistamicin biosynthesis that diffracted to a resolution of 2.6 Å (Supplementary Table 5, Fig. 5). Analysis of the separate protein domains showed that both the OxyA_{kis} enzyme[9,31–37] and the X-domain[9] maintained their anticipated folds, with little difference to comparable reported structures (Fig. 5–6; Supplementary Tables 6–7). OxyA_{kis} shows the archetypal P450 fold, which is largely composed of α-helices aside from one region of β-sheets[7]. The active site heme iron is coordinated by the thiolate of the conserved, proximal Cys332 residue, whilst the I-helix that runs across the top of the heme contains the

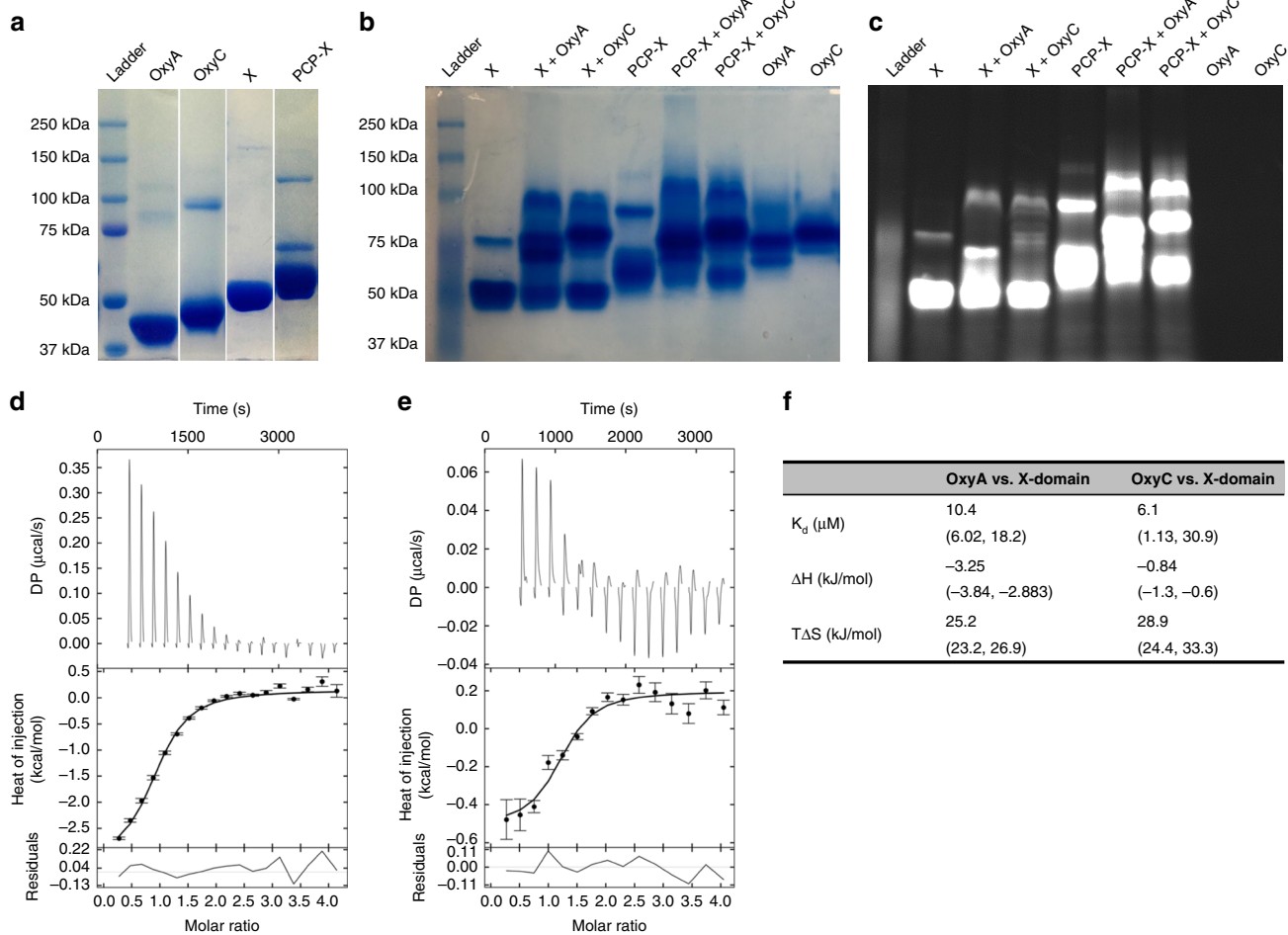

**Fig. 4** Interaction of OxyA$_{kis}$ and OxyC$_{kis}$ with the X-domain from the kistamicin NRPS. **a** SDS-PAGE of OxyA$_{kis}$ (44.9 kDa), OxyC$_{kis}$ (48.8 kDa), the X$_{kis}$-domain (52.0 kDa) and PCP-X$_{kis}$ (60.2 kDa). Native PAGE of the X$_{kis}$-domain and PCP-X$_{kis}$ didomain in isolation and after co-incubation with OxyA$_{kis}$ and OxyC$_{kis}$ visualised both with Coomassie blue stain (**b**) and using UV visualisation (**c**, following prior FITC labelling of the X$_{kis}$-domain and the PCP-X$_{kis}$ didomain). **d** Representative ITC isotherm data for interactions between (**d**) OxyA$_{kis}$ and (**e**) OxyC$_{kis}$ with the X$_{kis}$-domain. The upper panels represent baseline-corrected power traces; by convention, negative power corresponds to exothermic binding. The middle panels represent the integrated heat data fitted to the single binding sites model in SEDPHAT[58], with figures produced using GUSSI[59]. The bottom panels show the residues of the fit with error bars are standard error in the integration of the peaks as calculated by NITPIC ($n = 1$)[57]. **f** Binding affinities of OxyA$_{kis}$ and OxyC$_{kis}$ with the X$_{kis}$-domain. Affinities were determined by ITC at 25 °C with stirring at 300 rpm. 68.3% confidence intervals (1 Std. Dev.) are given in parentheses for $K_a$ and $K_d$ as calculated from a single titration. Source data are provided as a Source Data file

Glu/Gln pair of residues (Glu228, Gln229) that are required in P450 enzymes to maintain an effective protonation cascade during oxygen activation. The closest structural homologue to OxyA$_{kis}$ was identified as the related GPA cyclisation enzyme OxyA$_{tei}$ (RMSD 1.5 Å), which includes the same orientations of the capping F- and G-helices above the active site (Supplementary Fig. 32, Supplementary Table 6)[34]. The structure of OxyA$_{kis}$ reveals gaps in the observable density of several of the mobile regions surrounding the active site, which is unsurprising given the lack of substrate peptide bound to this P450 (Fig. 6).

The X$_{kis}$ domain within the OxyA$_{kis}$ complex adopts the typical C-domain fold composed of a pseudo-dimer of chloramphenicol acetyl transferase-like domains (Fig. 5–6). It is similar to the teicoplanin X-domain (solved in complex with OxyB$_{tei}$) in sequence (53.4%) and in structure, showing an RMSD of only 1.6 Å (Supplementary Fig. 32, Supplementary Table 7)[9]. The high structural similarity between these two X-domains stands in contrast to the much higher RMSD values seen with other condensation or epimerisation domains

(Supplementary Table 7)[38–43], which is typically due to differences in the relative orientations of the N- and C-terminal subdomains[6]. The only noticeable differences between the structures of X$_{kis}$ and X$_{tei}$ stem from some slight motions at the end of the loop from region encompassing residues 1399–1411 (visible in the OxyA$_{kis}$ complex, not in the OxyB$_{tei}$ complex) and differences in the turn between β-sheets (residues 1436–1441). The X$_{kis}$ domain shows the same mutated active site residues as seen in the X$_{tei}$ structure, which includes the mutation of crucial histidine and glycine residues (HHxxxDG to HRxxxDE) that render the X-domain inactive for condensation or epimerisation activity as well as blocking the access channel to the active site itself (Fig. 6)[9].

In terms of the interaction between the two proteins, the X$_{kis}$-domain remains closely associated to the PxDD motif at the start of the OxyA$_{kis}$ F-helix, but now the X-domain has rotated in towards the C-terminus of OxyA$_{kis}$, increasing the number of favourable protein-protein contacts around the OxyA$_{kis}$ D/E helices and connecting loop (Fig. 5). Burial of a substantial surface

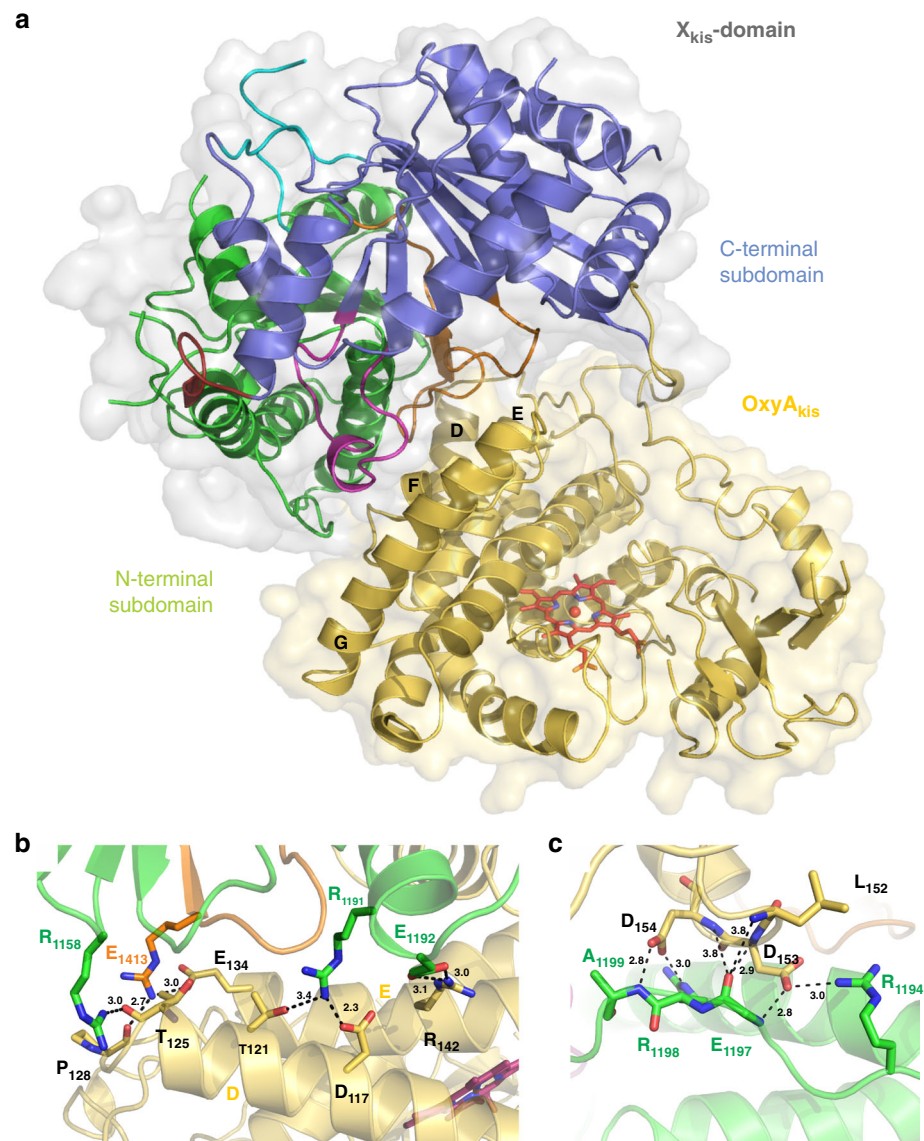

**Fig. 5** Structure of OxyA$_{kis}$/X$_{kis}$ complex. **a** Structure of the complex, showing the X-domain bound above the P450 via the D/E and F/G helices (X-domain N-terminal subdomain shown in green, C-terminal subdomain shown in blue, crossover regions shown in pink and orange; loops that close the typical acceptor site shown in cyan; P450 shown in yellow, with the active site heme moiety shown in sticks and helices D-G labelled). **b** Hydrogen bonding and salt bridge interactions between the X-domain and D-E helices of OxyA$_{kis}$ (coloured as in **a**, side chains shown as sticks). **c** Hydrogen bonding and salt bridge interactions between the X-domain and PxDD motif at the start of helix F in OxyA$_{kis}$ (coloured as in **a**, side chains shown as sticks)

area is observed in both complexes, consistent with the positive binding entropy, with the OxyA$_{kis}$/X$_{kis}$ complex displaying a slightly larger buried surface area than the OxyB$_{tei}$/X$_{tei}$ complex (976 Å$^2$ vs 725 Å$^2$, respectively; Supplementary Table 8)[9]. There are a number of H-bond and salt bridge interactions (17 and 20, respectively) in both complexes, which are likely to be important in controlling binding specificity[8,9]. The OxyA$_{kis}$/X$_{kis}$ complex is consistent with the ITC data and serves to confirm both the general binding mode of Oxy enzymes to X-domains during GPA biosynthesis and also the importance of the polar interactions in the interface, presumably to maintain specificity for Oxy/X binding[6].

**In vitro reconstitution of kistamicin Oxy enzyme activity.** To explore the cyclisation activity by the kistamicin Oxy enzymes in vitro, we first established that these enzymes maintained the essential heme thiolate ligation state following *E. coli* over-expression by obtaining reduced, carbon monoxide-bound UV/vis spectra showing the typical P450 absorbance of active enzymes, albeit with ~ 50% protonated, inactive P420 component (Supplementary Fig. 33)[7]. To explore the activity of OxyA$_{kis}$ and OxyC$_{kis}$, we synthesised a series of nine peptides based on the kistamicin sequence that varied in length from 3–7 amino acids. Structurally, these explored the effects of altering the stereochemistry of Hpg-4, and also the role of the stereochemistry/ structure of Dpg-3. All peptides were synthesised using conditions reported to avoid Hpg/Dpg racemisation[44,45]. Synthesis was performed on hydrazine resin to enable generation of the corresponding peptidyl CoAs following resin activation and thioester formation with CoA (Supplementary Figs. 34–43, Supplementary Table 9)[46]. Following purification, the peptides were loaded onto the PCP-X$_{kis}$ construct using the phosphopantetheinyl transferase Sfp from *B. subtilis*[47] and

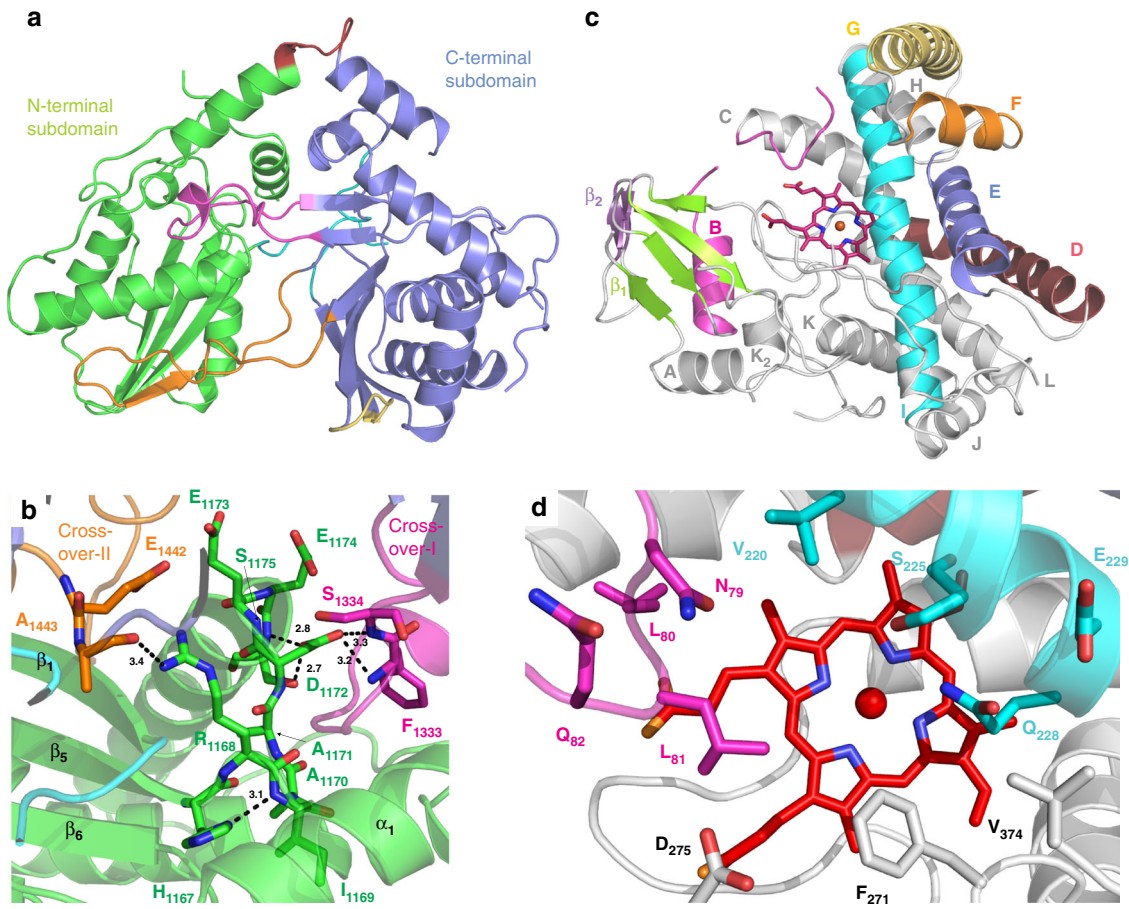

**Fig. 6** Structure of individual domains within the OxyA$_{kis}$/X$_{kis}$ complex. **a** Structure of the X-domain from kistamicin biosynthesis (coloured as in Fig. 5). **b** Typical active site region of a C/E-type domain showing the effects of mutating the conserved HHxxxDG motif into the HRxxxDE motif found in X$_{kis}$ (coloured as in Fig. 5, side chains shown as sticks). **c** Structure of the OxyA$_{kis}$ enzyme (B-helix and loop shown in magenta, D-helix shown in firebrick red, E-helix shown in blue, F-helix shown in orange, G-helix shown in yellow, I-helix shown in cyan, β-1 region shown in green, β-2 region shown in purple, heme shown in red sticks). **d** Active site of OxyA$_{kis}$ (coloured as in **c**, side chains shown as sticks)

subsequently incubated with the enzymes OxyA$_{kis}$, OxyC$_{kis}$ or OxyB$_{tei}$ (or a combination thereof) and appropriate redox system before cleavage of the thioesters with methylamine and analysis by LCMS (Fig. 7, Supplementary Dataset 1, Supplementary Figs. 44–52)[48]. OxyC$_{kis}$ proved to be highly active on tetrapeptides (K3–6, K4-7) that covered the C-O-D ring (positions 4–6 of the peptide) when Hpg-4 was present in the (D)-configuration and significantly more active than OxyB$_{tei}$ (Fig. 7c/e, Supplementary Figs. 45, 47, 49, 50). The alteration of Hpg-4 to an (L)-configuration reduced turnover activity to low levels (Supplementary Fig. 46), indicating the importance of the (D)-configured Hpg-4 residue for Oxy activity and supporting the (D)-configuration of this residue despite the lack of a distinct E-domain in module 4 of the kistamicin NRPS. In the case of the K3–7 pentapeptide, OxyC$_{kis}$ and OxyB$_{tei}$ were both highly active, with indications of a bicyclic product in the OxyC$_{kis}$ reactions (Fig. 7f, Supplementary Figs. 48, 53). OxyC$_{kis}$ and OxyB$_{tei}$ were both less active with longer peptide substrates K1-6 and K1-7 (Fig. 7g/i, Supplementary Figs. 49–50). The position of the C-O-D ring insertion performed by OxyC$_{kis}$ was confirmed by MS/MS for both K4–7 and K1-6 turnovers (Supplementary Figs. 54–55). Unexpectedly, OxyC$_{kis}$ also showed high cyclisation activity for the K1-4 tetrapeptide as a substrate, which was not observed for OxyB$_{tei}$ (Fig. 7h, Supplementary Fig. 51). MS/MS analysis of the monocyclic

products showed that there was a mixture of products present in this case, with products identified possessing linkages between Tyr-1/Hpg-3 and Tyr-1/Hpg-4 (but none involving Trp-2, Supplementary Fig. 56). Shorter tripeptide substrates K2-4 were very poor substrates for all enzymes tested here (Fig. 7d, Supplementary Fig. 52). Disappointingly, no cyclisation activity could be reconstituted for OxyA$_{kis}$, either in isolation of combination with other Oxy enzymes, despite extensive attempts.

Finally, we explored the tolerance of OxyC$_{kis}$ and a Type I homologue (OxyC$_{cep}$) for changes in the C-terminal residue of the peptide substrate. A teicoplanin-like peptide with a Hpg-7 residue (T1-7) was prepared as a substrate for an AB ring forming OxyC$_{cep}$ and P1-7 with a Dpg-7 residue found in all Type I–IV GPAs was used as a substrate for OxyC$_{kis}$ (Supplementary Fig. 57). Following cyclisation with OxyB and OxyA enzymes[46], incubation of the bicyclic T1-7 peptide with OxyC$_{cep}$ then demonstrated the ability of this enzyme to generate a tricyclic compound, whilst OxyC$_{kis}$ was unable to install a third ring in the bicyclic P1-7 peptide. We furthermore ascertained that OxyC$_{kis}$ alone or in combination with OxyB$_{tei}$ was unable to install a second crosslink into the T1-7 peptide (Supplementary Dataset 1). These results fit with the lack of complementation of Type I/IV OxyC deletion strains using the kistamicin OxyC homologue, which could well stem from alterations to the N-terminus of the

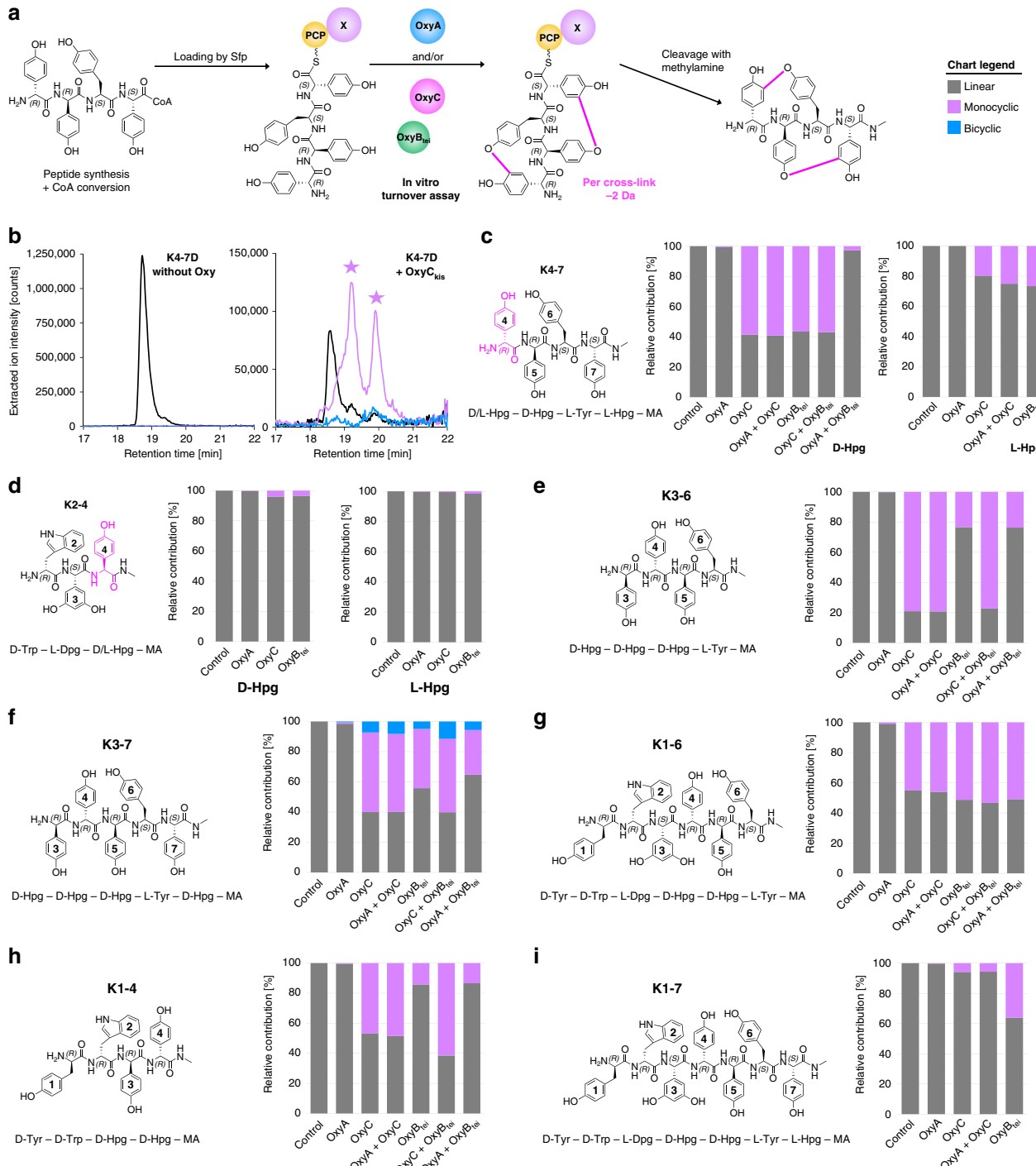

**Fig. 7** In vitro reconstitution of kistamicin Oxy enzymes. **a** Schematic illustration of in vitro reaction: kistamicin peptides were synthesised on hydrazine resin and converted into their CoA thioesters that were then loaded into the PCP-X_kis didomain by the phosphopantetheinyl transferase Sfp. After incubation with OxyA_kis, OxyC_kis and/or OxyB_tei (different combinations), the peptide was cleaved from the PCP by the addition of methylamine and subsequently analysed by HPLC-MS. **b** Example shown for the peptide K4-7D: in the control reaction, only linear peptide is detected. Reactions containing OxyC_kis led to the formation of several monocyclic products and traces of bicyclic compounds. Structures of synthesised peptide probes and turnover results of **c** K4-7 (4-D) (left) and (4-L) (right), **d** K2-4 (4-D) (left) and (4-L) (right), **e** K3-6, **f** K3-7, **g** K1-6, **h** K1-4 and **i** K1-7. Detection of linear peptide mass is indicted in dark grey, of monocyclic compound in purple and bicyclic compound in blue. Source data are provided as a Source Data file

bicyclic peptide sequence and stereochemistry in this case. Despite this, the turnover results obtained with OxyC_kis show that this is a promiscuous crosslinking catalyst for the insertion of crosslinks between amino acids with phenolic side chains in a variety of sequence contexts.

## Discussion

The biosynthesis of the glycopeptide antibiotics is a highly complex process, which is necessitated by the complex structure of these peptide antibiotics[1]. Arguably the most complicated portion of the biosynthetic pathway is the late stage peptide cyclisation cascade, mediated by multiple Oxy enzymes acting on

NRPS-bound peptides[8,30]. Whilst the majority of GPAs identified to date possess Type I–IV structures and concomitant antimicrobial activity[1], the divergent structure and antiviral activity of the Type V GPA kistamicin makes this an important system to address questions of peptide and carrier protein construct selectivity in the broader context of GPA biosynthesis. The structure of kistamicin, its antiviral activity and the biosynthetic machinery responsible for its production is significantly different from that of typical Type I–IV GPAs that inhibit bacterial cell wall biosynthesis. The overall strategy remains an NRPS produced peptide that is oxidatively cross-linked by the action of a variety of Oxy enzymes. However, the divergence of Type V GPAs is evident in both the details of the peptide producing NRPS (see Supplementary Discussion) and also the Oxy-mediated peptide cyclisation cascade. Arguably the most striking difference found in kistamicin biosynthesis when compared to Type I–IV GPAs is the presence of only two Oxy enzymes in the cluster and yet three oxidative crosslinks within the final kistamicin structure (Fig. 1), implying that one of the Oxy enzymes is inserting two crosslinks in kistamicin. Phylogenetic analyses of the enzymes show that the $OxyA_{kis}$ enzyme clusters closely with complestatin homologue and not with the OxyA enzymes found in Type I–IV GPAs (Supplementary Fig. 2). This aligns with the role of both of these enzymes in the insertion of the Trp-2/Hpg-4 aryl (DE) crosslink and is unique to these Type V GPAs. Perhaps more curiously, the second kistamicin Oxy enzyme clusters within the OxyC clade of the phylogenetic tree and not with the OxyB enzymes of Type I-IV systems. This is a most unexpected result, as OxyB enzymes are otherwise universally responsible for the insertion of the first ring link (the C-$O$-D ring) into GPAs[10,25,26]. This also suggested that the $OxyC_{kis}$ enzyme could be responsible for the A-$O$-B ring in kistamicin, as this is the same position of linkage (although a different product ring) as the AB ring produced by OxyC homologues in Type I–IV GPAs.

Using gene deletion and subsequent complementation experiments in kistamicin, balhimycin and A47934 producer strains[10,25], it was possible to show that the deletion of either of the genes encoding $OxyA_{kis}$ or $OxyC_{kis}$ resulted in a loss of production of kistamicin. With our ability to delete the separate Oxy enzymes from the producer strain, we were able to show that the $OxyC_{kis}$ enzyme appears to act first in the cyclisation cascade, with only linear peptides detected in the absence of this gene (Fig. 2). This result is in line with studies from the complestatin system that shares the C-$O$-D/DE rings present in kistamicin, and where the $OxyA_{com}$ enzyme there is responsible for the incorporation of the DE ring[26]. Thus, the removal of $OxyA_{kis}$ would be expected to—and does—result in accumulation of monocyclic peptides bearing the C-$O$-D crosslink between residues Tyr-6 and Hpg-4 (Fig. 2, Supplementary Figs. 18–23). This also matches the results from gene deletion experiments from balhimycin and A47934[10,25], although in these cases there is a distinct OxyB enzyme that installs this ring linkage. The detection of monocyclic hexapeptide in addition to monocyclic heptapeptide has also been found to occur in these systems, with recent results showing that this is due to the stalling of the NRPS leading to the unusual oxidation of the hexapeptide in these cases[28]. The lack of an A-$O$-B/C-$O$-D bicyclic peptide product in the $OxyA_{kis}$ deletion strain also indicates that this must occur subsequent to the insertion of the DE ring by $OxyA_{kis}$, maintaining the order of activity determined for all other GPAs despite the unusual nature of the kistamicin cyclisation cascade[10,25,26]. Whilst order of enzyme activity determined thus far ($OxyC_{kis}$ then $OxyA_{kis}$) does not allow for the unambiguous assignment of the enzyme responsible for A-$O$-B ring insertion due to the inability to assay this directly through deletion of the gene responsible for either

Oxy, the presence of a bicyclic hexapeptide in the $OxyA_{kis}$ deletion strain does indicate that $OxyC_{kis}$ is able to perform bicyclisation of peptides in vivo (Fig. 2, Supplementary Figs. 24–25) and is somewhat promiscuous with respect to substrate. Complementation of the balhimycin and A47934 oxyA or oxyC deletion strains with the respective kistamicin homologues led to no antibiotic production. Whilst unsurprising in the case of OxyA due to the different ring linkage and residues involved, the lack of complementation for OxyC shows that there is a distinct difference between these enzymes from Type I/IV and V systems, which matches the phylogenetic analysis of the OxyC enzymes (Supplementary Fig. 2) and in vitro turnover experiments. To assess whether the differences in OxyC activity could be due to the different ring sizes formed in the Type I–IV vs kistamicin systems, computational studies were also undertaken to determine the relative ring strains between the Type I–IV AB ring (12 membered ring) as opposed to the larger kistamicin A-$O$-B ring (15 membered ring) (Fig. 3). Synthetically, installation of the AB ring in Type I–IV GPAs is highly challenging due to the rigidity of this system together with the interlinked C-$O$-D ring. However, computational analysis indicated that whilst the kistamicin A-$O$-B ring might intuitively appear easier to form (due to its larger size) this was not the case, and the kistamicin A-$O$-B ring is in fact more strained than the typical 12-membered AB ring. This supports the hypothesis that differences in the activities shown by OxyC enzymes from Type I–IV GPA biosynthesis compared to the $OxyC_{kis}$ enzyme is in the chemistry of the ring linkage and is not directly related to ring size.

Whilst significant differences exist between the crosslinking cascades of kistamicin and the Type I–IV GPAs, the peptide cyclisation process remains reliant on the unique GPA P450 interaction domain, known as the X-domain, for Oxy recruitment to the NRPS-bound peptide substrate[8,9]. The interaction of the X-domain with the Oxy enzymes from kistamicin biosynthesis was demonstrated by different biochemical techniques, and most importantly using ITC (Fig. 4). These experiments show an apparent Oxy to X-domain interaction in the low micromolar range, which is in agreement with data for other GPAs obtained through different methods[9,30]. The use of ITC here was most valuable, as it also showed the interaction was driven by entropically driven and maintained the anticipated 1:1 stoichiometry of binding that had been postulated from previous single turnover experiments. The structure of the kistamicin OxyA/X-domain complex confirms that the interface between the X-domain and different Oxy enzymes remains conserved across divergent GPA biosynthetic machineries and for Oxy enzymes at different points within the cyclisation cascade (Fig. 5–6). This shows that only a single Oxy enzyme can be bound to the NRPS at any one time, and indicates that each Oxy must be replaced on the NRPS by the subsequent Oxy enzyme to enable catalysis[8]. This is in agreement with previous biochemical assays that supported a conserved X-domain/Oxy interface and has important mechanistic implications for peptide cyclisation[9,30]. Binding data obtained for the teicoplanin system indicates that the rate determining step in P450-mediated GPA cyclisation occurs after initial Oxy/X-domain complex formation, with the selection of the individual Oxy enzyme for the structure of the peptide likely the determinant for tight binding and hence maintaining an effective peptide cyclisation cascade[8]. The $OxyA_{kis}$/$X_{kis}$ complex serves to confirm both the general binding mode of Oxy enzymes to X-domains during GPA biosynthesis and the importance of polar interactions in ensuring binding specificity between the Oxy enzymes and the X-domain. Furthermore, it shows that despite the divergent structure and biosynthetic machinery of kistamicin the requirements for the Oxy-mediated peptide cyclisation cascade remain conserved across GPAs.

Whilst attempts to reconstitute both kistamicin Oxy enzymes was not successful due to the lack of any observable activity for OxyA$_{kis}$, the characterisation of OxyC$_{kis}$ activity against synthetic peptides showed that this enzyme was highly active (Fig. 7). Indeed, comparisons to the OxyB enzyme from the teicoplanin system (OxyB$_{tei}$)[35] showed that the activity of OxyC$_{kis}$ even exceeded the activity of OxyB$_{tei}$ when using short peptides (Supplementary Figs. 44–52). This was particularly the case for the cyclisation observed with the N-terminal K1-4 peptide and OxyC$_{kis}$, where high activity was detected and a mixture of products including 1/3 and 1/4 monocyclic peptides were detected (Supplementary Fig. 56). This result is in line with the appearance of different bicyclic peptides in the in vivo OxyA$_{kis}$ knockout, although interestingly in this case only ring linkages were detected to residues 2 and 3 of the peptide, likely due to the increased rigidity of the C-O-D crosslinked monocyclic peptide substrate. There were also indications that OxyC$_{kis}$ is capable of bicyclisation of peptides in in vitro experiments, albeit at low levels. This, however, is unsurprising given the in vivo gene disruption results, which suggest that OxyC$_{kis}$ activity is followed by OxyA$_{kis}$ mediated insertion of the DE ring. The promiscuity of the OxyC$_{kis}$ enzyme, its ability to perform bicyclisation both in vivo and in vitro, the common phenolic crosslinking chemistry employed by the Oxy enzymes and the phylogenetic analysis placing this enzyme in the OxyC group all support the activity of this enzyme as the insertion of both the C-O-D and A-O-B ring within kistamicin biosynthesis (Fig. 8).

The cyclisation process in GPA biosynthesis is an impressive feat of biological chemistry and one that appears to be largely conserved amongst the GPAs. This remains true even in the case of the Type V GPAs such as kistamicin, where the structure of these peptide natural products and their resultant activity profiles have changed significantly from those involved in the inhibition of bacterial cell wall biosynthesis. This conservation of mechanism includes the nature of the Oxy catalysts responsible, the timing of peptide cyclisation occurring on the NRPS-bound peptide and the mechanism of Oxy recruitment, which is mediated through the conserved X-domain found in the final NRPS module. However, the process of peptide cyclisation during kistamicin biosynthesis also shows several major differences to typical GPA oxidative cascades, of which the most significant is the ability of one Oxy enzyme to install two crosslinks within the peptide substrate. Our results strongly support the hypothesis that this role is performed by OxyC$_{kis}$, which is capable of installing not only the C-O-D type crosslink found in all GPAs but also further crosslinks within the same peptide. However, inclusion of an A-O-B ring in kistamicin in place of an AB ring—which is highly important for the antibiotic activity of Type I–IV GPAs—underlines the fact that the challenging nature of the GPA crosslinking cascade has led to the evolution of catalysts that fit the specific requirements of each type of biosynthetic system. Whilst we do not yet completely understand the role that kistamicin plays for the producing organism (although it and the related compound complestatin have been shown to possess antiviral activity), it is clear that the biogenic route for this compound has undergone a divergent optimisation process that leads to a dual-functional Oxy enzyme within the peptide cyclisation cascade. This is another example of the multitude of roles that P450 catalysts play in natural products biosynthesis and one that raises the prospects of producing further, modified GPA structures with divergent activity through biosynthetic redesign.

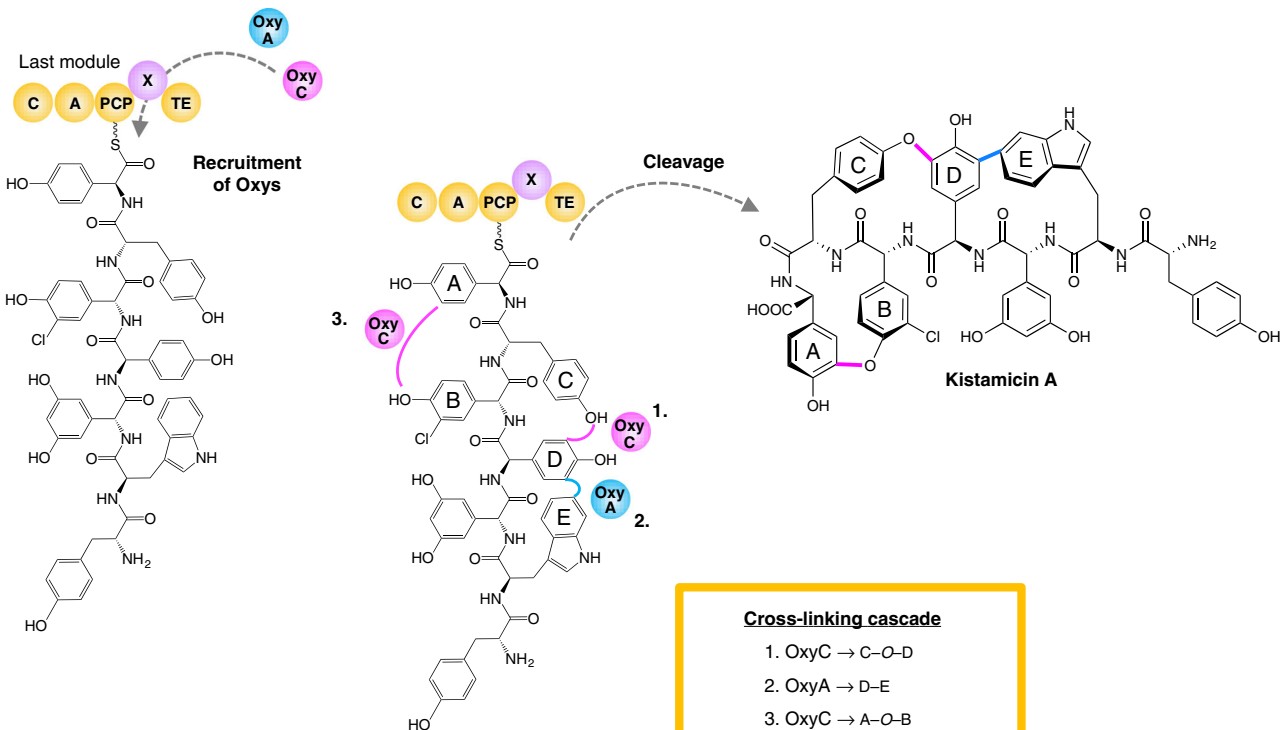

**Fig. 8** Postulated late-stage modification of the peptide in kistamicin biosynthesis. OxyA$_{kis}$ and OxyC$_{kis}$ are recruited to their NRPS-bound heptapeptide substrates by the X-domain present in the last module of the NRPS machinery. OxyC$_{kis}$ introduces the first crosslink—the C-O-D ring, which is followed by OxyA$_{kis}$-catalysed insertion of the D-E ring that is only present in Type V GPAs. Based on the reactivity and promiscuity of the OxyC$_{kis}$ enzyme, we then hypothesise that OxyC$_{kis}$ acts to install the third crosslink, the expanded A-O-B crosslinking, before the completed kistamicin A is cleaved from the NRPS by the actions of the terminal thioesterase domain

## Methods

**Chemicals and reagents**. Common reagents were obtained from Bachem, Sigma-Aldrich/ Merck, Ajax Finechem or Chem Supply. Coenzyme A was obtained from Affymetrix; glucose dehydrogenase was obtained from Sorachim.

**Cultivation and generation of *Actinomadura parvosata* mutants**. The bacterial strains and plasmids used for the generation of *A. parvosata* mutants are listed in Supplementary Table 2. *Escherichia coli* NovaBlue and NEBα were used for cloning purposes, and the methylation-deficient strain *E. coli* ET12567 pUB307 was used for intergeneric conjugation. Primers are listed in Supplementary Table 3.

*Actinomadura parvosata* subsp. *kistnae* (also known as *Nonomuraea* sp. ATCC 55076) is the kistamicin-producing wild-type and was used to generate the deletion mutants ΔkisN and ΔkisO.

*E. coli* strains were grown in Luria broth (LB) medium at 37 °C and were supplemented with 100 μg mL$^{-1}$ apramycin when necessary to maintain plasmids. Liquid cultures of *A. parvosata* were cultivated in 50 or 100 mL of R5 medium in 100 or 500 mL Erlenmeyer flasks with steel springs at 29 °C and 180 rpm for 3–10 days. Liquid/solid media were supplemented with 50 μg mL$^{-1}$ apramycin to select for strains carrying integrated antibiotic-resistance genes. Media are listed in Supplementary Table 4.

Gene deletions were made by implementation of the GUS (β-glucuronidase) selectable marker system to construct the *kisN* and *kisO* gene deletion plasmids, containing the upstream and the downstream flanking regions of *kisN* and *kisO*, fragments with sizes of 1.5 kb each, were amplified by PCR using the genomic DNA of *A. parvosata* as template and the primers KisN-down Fw, KisN-down Rv, KisN-up Fw, KisN-up Rv, KisO-down Fw, KisO-down Rv, KisO-up Fw, KisO-up Rv containing *Xba*I/*Nde*I and *Nde*I/*Hind*III restriction sites at the 3′ and 5′ ends). The PCR amplicons KisN-down, KisN-up, KisO-down and KisO-up were separately introduced into the pJET blunt vector yielding pJET_KisN-down, pJET_KisN-up, pJET_KisO-down and pJET_KisO-up. For the construction of the deletion vector pGUSA21_KisN-up_down and pGUSA21_KisO-up_down, the fragments KisN-down, KisN-up, KisO-down and KisO-up were excised from pJET with *Xba*I/*Nde*I for KisN-down, KisO-down and *Nde*I/*Hind*III for KisN-up, KisO-up and cloned into pGUSA21 yielding pGUSA21_KisN-up_down, and pGUSA21_KisO-up_down. The final plasmids were transferred into *E. coli* ET12567 pUB307 and finally into *A. parvosata* by intergeneric conjugation. The integration of the pGUS-deletion plasmids encoding β-glucuronidase (Gus) gene was determined based on blue-white screening, by plating the trans conjugants on R5 agar containing 20 mM X-gluc (5-bromo-4-chloro-1H-indol-3-yl ß-D-glucuronic acid). PCR approach was used to verify the single crossover events using the primer pairs pGUS-FP/pGUS-RP.

The obtained single crossover mutant strains *A. parvosata*_pGUSA21_KisN-up_down and *A. parvosata*_pGUSA21_KisO-up_down were used for the generation of the in-frame deletion mutants *A. parvosata* ΔkisN and *A. parvosata* ΔkisO (Supplementary Fig. 12). In order to select for *A. parvosata* ΔkisN and *A. parvosata* ΔkisO gene deletion mutants, in which a double crossover event via the second cloned fragment had occurred, single mutants of strains *A. parvosata*_pGUS KisN-up_down and *A. parvosata*_pGUS KisO-up_down were cultivated for two days in 50 ml R5 medium at 29 °C and 180 rpm under apramycin selection. Afterwards the mycelium was washed and used for inoculation of 50 mL fresh R5 medium without antibiotic selection and cultivated by 37 °C and 180 rpm for 24 h. The cultures were then centrifuged. The mycelium was fragmented by sonication and the protoplasts were prepared as described by Matsushima and Baltz[49]. Diluted protoplasts were plated on R5 agar plates overlaid with 20 mM X-gluc and red colonies (Supplementary Fig. 13) were picked for control PCRs with three primer pairs ΔKisN Fw/ΔKisN Rv, kisN-NdeI-Fw/ΔKisO Rv for KisN and ΔKisO Fw/ΔKisO Rv, KisN Fw/ kisO-XbaI-Rv for KisO (Supplementary Figs. 14, 16).

For ΔkisN and ΔkisO complementation, the expression plasmids pRM4.2[50] was used to construct pRM4_kisN and pRM4_kisO. The primer kisN-NdeI-Fw including with 5′ *Nde*I and kisN-HindIII-Rv with 3′ *Hind*III restriction site were used to amplify the complete *kisN* gene from genomic DNA of *A. parvosata*. The primer kisO-HindIII-Fw carrying a 5′ *Hind*III site and kisO-XbaI-Rv with 3′ *Xba*I restriction site were used to amplify the complete *kisO* gene from genomic DNA of *A. parvosata*. The amplified fragments of 1172 bp for *kisN* and 1301 bp for *kisO* were inserted downstream of the *ermE** promoter of the pRM4 vector, respectively, resulting in the plasmid pRM4_kisN and pRM4_kisO. The complementation plasmids were introduced into *A. parvosata* using a standard protocol for intergeneric conjugation in actinomycetes, where upon they integrated at the chromosomal φC31 *attB* sites of *A. parvosata*. Recombinant mutants containing the integrated pRM4_kisN plasmid were selected on apramycin plates and confirmed by PCR (Supplementary Fig. 15) using the primers kisN-NdeI-Fw/ kisN-HindIII-Rv and pRM4SeqRv/ pRM4SeqFw).

**Production of kistamicin in *A. parvosata* mutants**. Wildtype *A. parvosata*, the deletion mutants *A. parvosata* ΔkisN and *A. parvosata* ΔkisO and the complemented mutants were cultivated for 7 days on petri dishes containing 30 mL of SFM-agar) for sporulation. For the fermentations, 5 mL of three days-old preculture (50 μL spores grown in 50 ml R5 medium) were used for inoculation of 100 mL R5 medium. Fermentations were carried out for seven to ten days at 29 °C in a rotary shaker at 180 rpm. Cultures were centrifuged and the pellets were

extracted with methanol for 30 min and dried by rotary evaporation (Heidolph laborota 4000). Dried extracts were resuspended in 500 μL methanol. Consequently, 200 μL of chloroform was added and carefully mixed. After addition of 200 μL of water, the samples were centrifuged at 20,000xg for 5 min. The peptides in the upper layer were further purified by solid phase extraction (Strata-X polymeric reversed phase (Phenomenex)) and dried by speed evaporation. The extract got dissolved by H$_2$O + 0.1% FA for HRMS and MS/MS analysis.

**HRMS and MS/MS analyses**. Samples in up to 20% acetonitrile were separated on a RSLC 3000 LC system (Thermo) coupled to an Orbitrap Fusion Tribrid mass spectrometer (Thermo Scientific). The LC system consisted of a trap column Acclaim PepMap 100 (100 μm × 2 cm, nanoViper, C18, 5 μm, 100 Å; Thermo Scientific) and an Acclaim PepMap RSLC analytical column (75 μm × 50 cm, nanoViper, C18, 2 μm, 100 Å; Thermo Scientific). Samples were loaded onto the trap column in μL-pickup mode using 2% acetonitrile, 0.1% TFA transport liquid. The columns were developed with a 30 min gradient from 6 to 30% acetonitrile in 0.1% formic acid at 250 nL/min coupled to the mass spectrometer nanospray source operated at 1.7 kV. The mass spectrometer was operated in either data dependent, PRM and/or SIM mode to target the appropriate species. Full scans were acquired at 70–500 k resolution and MS2 spectra acquired at 35 k or 70 k resolution, with a 1.5-1.8 m/z isolation window and normalised collision energy between 25 and 33.

Raw data was manually analysed in XCalibur QualBrowser (Thermo Scientific), with extracted ion chromatograms to the predicted species routinely generated with 10 ppm mass tolerance. MS$^2$ spectra corresponding to the predicted mass were manually characterized for ring closures based on predicted peaks calculated using MSProduct from the Protein Prospector suite (allowing for different ring linkages) and comparing differential peaks between related spectra. Data was also analysed in Skyline (v4.2.0, University of Washington), using both small molecule and manually curated modified peptide workflows for predicted species with parent ions extracted for five isotopes plus the M-1 isotope to confirm the correct number of rings. PRM data was examined for multiple peaks corresponding to different ring closure forms and MS$^2$ fragment profiles across chromatographic peaks were studied for heterogeneous ring closures.

**Extraction and purification of kistamicin A**. *A. parvosata* was cultivated in 10 × 100 mL R5 medium[51] in an orbital shaker (220 rpm) 500 mL baffled Erlenmeyer flasks with one baffle and steel springs at 27 °C. The flasks were inoculated with 2% (v/v) of 2 days old preculture. After 7 days of fermentation, the cells were collected by centrifugation. The kistamicin containing mycelium was extracted three times with 500 mL acetone. This extract was concentrated under reduced pressure to remove acetone. The aqueous residue was adjusted to pH 8–9 and extracted three times with 300 mL butanol. The aqueous layer was discarded and the combined organic layers were concentrated to dryness in vacuo. The dried extract was dissolved in dichloromethane and applied to a LiChroprep Diol column (particle size 40–63 μm, 25 mm × 300 mm). The metabolites were eluted by a stepwise gradient of dichloromethane/methanol (98:2, 95:5, 9:1, 8:2, 7:3, 5:5, respectively) to afford 17.6 mg of kistamicin (chemical formula C$_{61}$H$_{51}$ClN$_8$O$_{15}$; molecular weight 1171.5; m/z [M + H]$^+$ 1172.5, [M-H]$^-$ 1170.5).

**Characterisation of kistamicin A**. HRMS spectra were obtained on Triple time of flight MS system in both positive and negative modes. The sample (dissolved in milliQ water) was separated using reversed-phase chromatography on a Shimadzu Prominence nanoLC system (sample desalted on an Agilent C18 trap (0.3 × 5 mm, 5 μm) for 3 min using a flow rate of 30 μL/min, followed by separation on a Vydac Everest C$_{18}$ (300 Å, 5 μm, 150 mm × 150 μm) column at a flow rate of 1 μL/min). A gradient of 10–60% buffer B was then applied over 30 min where buffer A = 1% MeCN/0.1% formic acid and buffer B = 80% MeCN/0.1% formic acids was used to separate peptides. Eluted peptides were directly analysed on a TripleTof 5600 instrument (ABSciex) using a Nanospray III interface. Gas and voltage settings were adjusted as required. MS TOF scan across m/z 200–2400 was performed for 0.5 sec followed by information dependent acquisition of up to 10 peptides across m/z 40–2400 (0.05 sec per spectra). HRMS calculated for C$_{61}$H$_{50}$ClN$_8$O$_{15}$ [M-H]$^-$ 1169.3090, observed 1169.3159.

NMR spectra were recorded at 298 K on a Bruker Avance 700 spectrometer. $^1$H and $^{13}$C NMR spectra were recorded at 700 MHz and 175 MHz, respectively, with the residual protonated signal in the methanol-d$_4$ (δ$_H$ 3.31), the central peak of the methanol- d$_4$ septet (δ$_C$ 49.0) and TMS as internal standards. The NMR data (Supplementary Figs. 4–11) is in excellent agreement with the reported data for kistamicin A[18], and indicates the presence of sucrose within this sample. However, Naruse et al. reported the NMR assignments for kistamicin A in methanol-d$_4$ (proton) and in DMSO-d$_6$ (carbon). We report all the NMR assignments in methanol-d$_4$ based on HMBC and HSQC spectra (Supplementary Table 1). We also recorded $^1$H and $^{13}$C NMR spectra in DMSO-d$_6$, but they were too complex for simple interpretation and consisted of many overlapping signals. We also reverse here the previously published assignments for B-2 and B-6 hydrogens of kistamicin A on the basis of the HSQC and HMBC spectra obtained here. The assignments for B-2 and B-6 were established and confirmed by the presence of

HMBC correlations from $\delta_H$ 5.55 (B-2) to $\delta_C$ 132.5 (B-3) and $\delta_C$ 140.3 (B-4), and also from $\delta_H$ 5.70 (B-6) to $\delta_C$ 140.3 (B-4) and $\delta_C$ 150.7 (B-5).

**Genome sequencing and identification of _kis_ gene cluster**. The genomic DNA was isolated according the instruction of the bacterial DNA kit (VWR, Life Science, Belgium). In order to lyse the cells, 20 mg mL$^{-1}$ lysozyme was added during this procedure. Sequencing was performed by LGC Genomic GmbH using Illumina MiSeq V3; the genome was subsequently assembled by LGC Genomic GmbH. Prediction of the gene clusters was performed using antiSMASH 3.0[52]. The sequence of the kistamicin biosynthetic gene cluster was further analysed and annotated using BLAST (Supplementary Fig. 1).

**Phylogenetic analysis and alignment of proteins**. The protein sequences of the Oxy enzymes and the C-domains from the kistamicin biosynthetic gene cluster were compared with those from other GPA biosynthetic machineries by first aligning the protein sequences were using Muscle[53] and subsequent manual curation. Maximum likelihood phylogenies were created using RaxML[54] with 100 bootstrap replicates as a measure of branch support (Supplementary Figs. 2–3).

**Modelling of ring strain in alternate GPA structures**. For complete methods and detailed discussion please see Supplementary Information and Supplementary Figs. 27–31.

**Heterologous expression and purification of proteins**. The genes for OxyA$_{kis}$ (UniProt ID: A0A1V0AG67), OxyC$_{kis}$ (UniProt ID: A0A1V0AGL8) and the PCP-X didomain (residues 944–1493, UniProt ID: A0A1V0AGC2) were obtained as synthetic genes that had been codon optimised for _E. coli_ expression (Eurofins Genomics, Germany). The Oxy genes were designed to contain a 5′-_Nde_I and 3′-_Hind_III restriction site to clone them into pET28a expression vector (Novagen), respectively. The constructs possess a N-terminal His$_6$-tag. The PCP-X construct was designed with 5′-_Nco_I and 3′-_Xho_I restriction sites and cloned into the pET28-MBP1d vector that results in the expression of the protein as a N-terminal MBP fusion protein to improve protein yield (MBP-PCP-X). Furthermore, the standalone X domain ((residues 1024–1493, UniProt ID: A0A1V0AGC2)) was amplified from the optimised gene encoding the PCP-X protein using X-for and the T7$_{rev}$ primer and after _Nco_I/_Xho_I restriction also cloned into pET28-MBP-1d (MBP-X). The MBP expression enhancing tag with a N-terminal His$_6$-tag can be cleaved by TEV protease[29] They further possess a C-terminal Strep-II tag. All constructs were verified by sequencing using standard T7 primers.

_OxyA$_{kis}$_, _oxyC$_{kis}$_, _mbp-pcp-x_ and _mbp-x_ were expressed in _E. coli_ ArcticExpress (Agilent Technologies). Therefore, 1% (v/v) of an overnight culture with the expression vector was used to inoculate 10 L of autoinduction media ZYM-50524[55] supplemented with kanamycin (50 mg L$^{-1}$). For Oxy expression, the media was furthermore supplemented with the heme-precursor ∂-aminolevulinic acid (Carbolutions Chemicals GmbH) (0.1 g L$^{-1}$). The cultures were incubated for 5 h at 37 °C. The temperature was reduced to 18 °C for 65 h.

Purification of the enzymes were performed by Ni$^{2+}$-NTA affinity, ion exchange and size exclusion chromatography using an ÄKTA system (GE Healthcare). After Ni$^{2+}$-NTA affinity chromatography in Tris-HCl (50 mM, pH 7.4), NaCl (200 mM) and imidazole (10–300 mM), the buffer was exchanged for anion exchange chromatography with Tris-HCl (50 mM, pH 8), NaCl (50 mM). As final step, gel filtration using a superose 12 column (vol 320 mL) in Tris-HCl (50 mM, pH 7.8), NaCl (200 mM) was performed.

The MBP expression enhancing tag was cleaved from MBP-PCP-X and MBP-X following initial Ni$^{2+}$-NTA affinity chromatography by the addition of TEV protease (250 μg mL$^{-1}$). After incubation for 3 h at RT, a second Ni$^{2+}$-NTA affinity chromatography was performed to separate the X domain (X) and the PCP-X didomain (PCP-X) from the His$_6$-tagged MBP.

**Interaction studies of Oxy enzymes**. For FITC-labelling of X and PCP-X, fluorescein isothiocyanate (FITC) (Sigma) was dissolved in DMSO (1 mg mL$^{-1}$). The X domain and the PCP-X didomain was dialysed in 0.1 M sodium carbonate (pH 9) (2 mg mL$^{-1}$) and 100 μL FITC was added to 1 mL of protein solution. The mixture was incubated ON at 4 °C (protect from light). Remaining FITC was quenched by the addition of ammonium chloride (50 mM) for 2 h and washed by the usage of Amicon Ultra centrifugal filters (Millipore, 10 kDa MWCO, 0.5 mL). The degree of labelling was calculated by $A_{494nm}/\varepsilon_{FITC} * M_{protein}$ and was assessed as > 99%. The Oxy$_{kis}$ enzymes (30 μM) were mixed with the labelled X domain or PCP-X (30 μM) in 100 μL Tris-HCl (50 mM, pH 7.2), NaCl (50 mM), respectively, and incubated ON at 4 °C (protect from light). Possible interactions were analysed by native PAGE. After electrophoresis, the gel was visualized by UV absorption. Following, the gel was stained using brilliant blue and the protein bands were visualised.

All ITC experiments were performed on a Nano-ITC low-volume calorimeter (TA Instruments). ITC experiments were performed at 25 °C with stirring at 300 rpm. Protein and ligand solutions were prepared in matched ITC buffer (50 mM phosphate buffer pH 7.4, 150 mM NaCl, 5% v/v glycerol) and thoroughly degassed before use. Titrations were performed using 300–400 μM OxyA/OxyC, and involved 1 × 1 μL, followed by 14 × 3 μL or 18 × 2.5 μL injections of 1.27 mM

X-domain. Data were analysed using NITPIC[56], SEDPHAT[57] and GUSSI;[58] the baseline-subtracted power was integrated, and the integrated heats were fit to the single binding site model (A + B ↔ AB hetero-association) model to obtain the association constant ($K_a$). Fitting was achieved by iteratively cycling between Marquardt-Levenberg and Simplex algorithms in SEDPHAT until modelling parameters converged. Data represents individual titrations ($n = 1$). In all, 68.3% confidence intervals (one standard deviation) were calculated using the automatic confidence interval search with projection method using F-statistics in SEDPHAT.

**Crystallisation of the OxyA$_{kis}$/X$_{kis}$ complex**. OxyA$_{kis}$ and the standalone X$_{kis}$ domain were mixed to a final concentration of 20 mg/mL (1:1) in Tris-HCl (50 mM, pH 7.8) with NaCl (200 mM). Initial screening was performed at the Monash Molecular Crystallisation Facility (MMCF) with subsequent optimization performed by hand in 48-well sitting-drop plates. The complex was crystallised using a sitting-drop vapour diffusion method by mixing 1 μL protein with 1 μL of an optimized reservoir solution containing Bis-Tris (0.1 M, pH 6.5), NaCl (300 mM) and PEG 3350 (30% v/v). Crystals (cubes ~200 μM) formed after one week at 20 °C. Crystals were cryoprotected by transfer in a drop made of the reservoir solution supplemented with glycerol (30% final concentration, v/v). Crystal were collected in cryoloops and flash frozen in liquid nitrogen. Data were collected at the Australian Synchrotron (Clayton, Victoria, Australia) on beamline MX1 at 100°K (Supplementary Table 5). Data processing was performed using XDS[59] and AIMLESS as implemented in CCP4[60]. Phases were obtained in a molecular replacement experiment using PHENIX in-built Phaser module[61] and with a model generated by PHYRE[62]. The structure was built and refined using COOT[63] for model building and PHENIX-refine for refinement[61]. All graphics were generated with Pymol (Schrödinger LLC). Analysis of the interface between OxyA$_{kis}$ and X$_{kis}$ was performed using PISA[64].

**CO difference spectra of Oxy enzymes**. Spectra were obtained using a Jasco V-750 spectrophotometer at 30 °C. The enzymes were diluted to 2.5 μM in Tris-HCl (50 mM, pH 7.4), reduced using 10 μL of a saturated solution of sodium dithionate (Sigma) and CO was generated by the addition of a small spatula of sodium boranocarbonate (Dalton Pharma Services). The UV/vis spectra were measured between 390–600 nm (Supplementary Fig. 33).

**Synthesis of peptide substrates**. Solid phase peptide synthesis was performed manually on 2-chlorotrityl chloride resin (scale 0.05 mmol, 200 mg). Resin swelling was performed in DCM (8 mL, 30 min), followed by washing with DMF (3x), treatment with 5% hydrazide solution in DMF (6 mL, 2 × 30 min), washing with DMF and capping with a solution of DMF/TEA/MeOH (7:2:1) (4 mL, 15 min). The first amino acid coupling used Fmoc-amino acid (0.06 mmol), COMU (0.06 mmol) and 2,6-lutidine (0.06 mmol, 0.12 M) and was performed overnight; a second coupling step was always accomplished to cap unreacted hydrazide groups using Boc-glycine-OH (0.15 mmol), COMU (0.15 mmol) and 2,6-lutidine (0.15 mmol, 0.12 M) for 1 h. For Fmoc-deprotection, a 1% DBU solution in DMF was used (3 mL, 3 × 30 s). Subsequent Fmoc amino acid coupling was achieved by activating Fmoc-amino acid (0.15 mmol) with COMU (0.15 mmol) and 2,6-lutidine (0.15 mmol, 0.12 M) for 40 min. This Fmoc removal and coupling cycle was repeated until the last amino acid that was introduced with a Boc protecting group. The hydrazide peptide intermediate was cleaved from the resin, including ᵗBu and Boc removal, using a TFA cleavage mixture (TFA/TIS/H$_2$O, 95:2.5:2.5 v/v′/v″, 5 mL) for 1 h with shaking at room temperature. The solution was concentrated under nitrogen stream to ∼1 mL and precipitated with ice cold diethyl ether (∼8 mL), followed by centrifugation in a flame-resistant centrifuge (Spintron) and washed three times with 5 mL of cold diethyl ether. All crude hydrazide peptides were purified using a preparative RP-HPLC, and purified hydrazide peptides subsequently converted to CoA-linked peptides. To achieve this, the peptide hydrazide (1–5 mM) was dissolved in buffer A containing urea (6 M) and NaH$_2$PO$_4$ (0.2 M), pH 3 (obtained via addition of HCl) and the reaction mixture was cooled to −15 °C using a salt/ice bath. Peptides 5 and 6 required initial solubilisation with DMSO to obtain a final concentration of 10–50 μM (1% v/v). In the next step, 0.5 M NaNO$_2$ (0.95 eq.) was added to the solution and stirred for 10 min before addition of coenzyme A (1.2 eq., dissolved in buffer A). The solution was adjusted to pH 6.5 by adding KH$_2$PO$_4$/K$_2$HPO$_4$ buffer (6:94 v/v 1 M, pH 8.0) and stirred for further 30 min on ice with monitoring by LCMS. Final CoA-peptides were purified using preparative RP-HPLC (gradient 10–40% ACN or 15–45% ACN in 30 min).

For analysis and purification, a HPLC-MS system from Shimadzu (LCMS-2020) was used. UV-spectra were recorded via a SPD-20A Prominence Photo Diode Array Detector in analytical mode and via a SPD-M20A Prominence Photo Diode Array Detector in preparative mode. Solvents employed were water 0.1% FA and ACN + 0.1% FA for analytical measurements and water + 0.1% TFA and ACN + 0.1% TFA for preparative runs. Turnover analyses were performed using a Waters XBridge® Peptide BEH C18 column, 300 Å, 3.5 μm, 4.6 mm × 250 mm employing a gradient of 5–95% ACN + 0.1% FA in 30 min. Crude peptides were purified using a preparative HPLC Waters XBridge® Peptide BEH C18 OBD™ prep column, 300 Å, 5 μm, 19 mm × 150 mm employing a gradient of 10–40% or 15–45% ACN + 0.1% FA in 30 min. See Supplementary Figs. 34–43, Supplementary Table 9.

**In vitro Cytochrome P450 activity assays**. The cyclisation activity of OxyA$_{kis}$ and OxyC$_{kis}$ was tested against various peptide substrates using a coupled enzyme assay. Different peptidyl-CoAs (120 μM) were loaded onto the PCP-X didomain (60 μM) using the phosphopantetheinyl transferase Sfp (6 μM) in HEPES (50 mM, pH 7.0), MgCl$_2$ (10 mM) and NaCl (50 mM) in a final volume of 87.5 μL. After incubation of 1 h at 30˚C, the unloaded peptidyl-CoAs were removed by the usage of centricon units (10 kDa MWCO) and 5x washing with 300 μL HEPES (50 mM, pH 7.0), NaCl (50 mM) at 20,000 g, 5 min, 4 °C. The loaded PCP-peptide (50 μM) was incubated with one or more Oxy enzymes (2 μM) and redox partners PuxB variant A105V (5 μM) and PuR (1 μM) in HEPES (50 mM, pH 7.0), NaCl (50 mM) at a final volume of 105 μL[48]. The reaction was initiated by the addition of NADH (2 mM), which was continually regenerated by glucose (0.33% w/v) and glucose dehydrogenase (9 U/mL). Reactions were incubated for 3 h at 30 °C. After, the peptides were cleaved from the PCP-domain by the addition of methylamine (0.5 M) with incubated for 15 min at RT, which resulted in the formation of methylamide peptides. The reactions were neutralised using dilute formic acid and the peptides were purified by solid phase extraction (Strata-X polymeric reversed phase (Phenomenex) and analysed by analytical HPLC-MS using a Shimadzu LCMS-2020 system. The turnover reactions were analysed on a Waters XBridge BEH 300 C$_{18}$ column (5 μM, 4.6 × 250 mm) at a flow rate of 0.8 mL min$^{-1}$ using the following gradient: 0–5 min 5% solvent B, 5–40 min up to 75% solvent B (solvent A: water + 0.1% formic acid: solvent B: acetonitrile + 0.1% formic acid). The peptides were analysed in positive mode $[M + H]^+$; a mass difference of 2 Da was expected per crosslink inserted (Supplementary Figs. 44–52, Supplementary Table 9).

**Reporting summary**. Further information on research design is available in the Nature Research Reporting Summary linked to this article.

## Data availability

The authors declare that all other data supporting the findings of this study are available within the paper and its supplementary information files. The raw data underlying Figs. 3, 4 and 7, as well as Supplementary Figs. 27–30, 45–52 and 57 are provided as Source Data files. All data underlying the results of this study are available from the authors upon reasonable request. The structure of OxyA-X has been deposited in the Protein Data Bank (PDB) on the 20th of August 2018 with the primary accession code 6M7L.

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

## Acknowledgements

G. Stier (BZH-Heidelberg) for fusion protein vectors; S. Bell (University of Adelaide) for redox proteins; J. Yin (University of Chicago) for the R4-4 Sfp expression plasmid; C. Brieke (MPI) for assistance with peptide synthesis; D. Maksel & G. Kong (MMCF) for assistance with crystal screening experiments; S. Stockert (University of Tübingen) for DNA isolation and kistamicin purification; T. Hackl (MPI-Hd) for assistance with sequence reconstruction. We would also like to thank the MX1 beamline scientists at the Australian Synchrotron for their support during data collection. Computational facilities were provided by the Australian Government through the National Computational Infrastructure's National Facility under the National Computational Merit Allocation Scheme. This work was supported by the Deutsche Forschungsgemeinschaft (Emmy −Noether Program, CR 392/1-1 (M.J.C); SFB766 program TP-A03 (E.S. and A.K.)); Monash University, EMBL Australia and the National Health and Medical Research Council (APP1140619 to (M.J.C.)); the Universities Australia/ DAAD 2016 Australia—Germany Joint Research Co-operation Scheme (Award ID 16679401) awarded to E.S. and M.J.C., the University of Queensland (Strategic Research Fellowship to E.H.K.) and further supported under Australian Research Council's Discovery Projects funding scheme (project number DP170102220 to M.J.C. and J.J.D.V., project numbers FT120100632 and DP180103047 to E.H.K.).

## Author contributions

M.J.C. designed the study. A.G. cloned constructs, expressed proteins and performed turnovers, binding assays and interactions studies. J.T., M.S. and Y.Z. synthesised and purified peptidyl-CoA substrates. T.I and A.G. performed crystallisation experiments; T.I solved and built the structure of the OxyA/X-domain complex; M.J.C. analysed the structure. D.I. and E.S. conducted in vivo deletion experiments. A.K. and E.S. isolated kistamicin and genomic DNA to allow determination of the kistamicin gene cluster. M.S. and M.P. assisted with turnover studies using peptides **T1-7** and **P1-7**. I.A. and J.J.D.V. performed the structural analyses of kistamicin. A.G, M.A., N.Z. M.J.C. and E.S. analysed the kistamicin gene cluster. E.H.K. and J.J.D.V. performed the computational analysis of ring strain. R.J.A.G and R.B.S. performed and analysed HRMS and MS/MS experiments. J.A.K. and C.J.J. conduced and analysed ITC experiments. M.J.C wrote the manuscript with contributions from A.G., J.J. D.V., E.S., N.Z. and E.H.K. Figures were prepared by A.G., N.Z., E.H.K. and M.J.C.

## Additional information

**Competing interests:** The authors declare no competing interests.

