## [Peer Review File · Nature Communications]

Reviewers' comments:

Reviewer #1 (Remarks to the Author):

The manuscript by Cryle and colleagues continues their structural and functional investigation of the P450 enzymes that are involved in cross-link formation between aromatic residues of the glycopeptide antibiotics. Previously, the authors have studied primarily teicoplanin biosynthesis and determined the order of cross-link formation, the structures of several enzymes involved and the rather interesting presence of a non-functional condensation domain variant called the X-domain that serves as an protein-protein interface to capture the P450s and direct them towards the substrate.

The current paper continues this work in a different system that is responsible for the production of kistamicin. There are several unusual features of this NRPS pathway, some of which are experimentally explored herein. The authors present a crystal structure of the OxyA P450 bound to the X-domain. This is the first OxyA type protein (unlike the OxyB of the teichoplanin in earlier work) crystallized bound to the X-domain. This confirms a similar interaction interface suggesting only one P450 can bind the X-domain at a time. Functional analysis then investigates an outstanding question in kistamicin biosynthesis, namely the presence of three cross-links but only two P450 proteins.

The paper requires significant revision as noted below. The structural determination and analysis is sound. However some of the introductory material is speculative and, as currently written, runs the risk of becoming accepted as proven without experimental evidence. Further, some of the protein interaction and functional analysis does not fully support the conclusions drawn. Finally, the paper would benefit from significant revision to improve clarity. These issues are described below.

Significant Issues

1. The presentation of the potential activation of the Tyrosine residue by a Tyr-RS is speculative but presented as conclusive, as is its eperimization by the condensation domain. Further, the lack of an E-domain in module 4 is rationalized with a dual-function C/E domain downstream. These are interesting variations but should not be presented in "Results" until they are experimentally addressed.
2. The authors examine a functional interaction between OxyA and OxyC P450s and various protein constructs. The gel shifts in Figure S12 do not support the conclusion, particularly with OxyA, of a functional interaction. The shifted band (arrow in OxyA + PCP-X) matches the band in the OxyA lane alone. This analysis is challenged by the ladder of bands from oligomers. Perhaps if this experiment had been run with much less X or PCP-X, the disappearance of the band for the unshifted protein would be more convincing. Similarly, the long arrows in Figure S13 (right panel) for the chemical cross-linking are likely to point to large aggregates of protein that failed to enter the running gel.
3. The functional analyses of Figures 7 and 8 are challenging. The authors attempt to demonstrate cross-linking being catalyzed by the OxyC protein. They conclude that they have shown this however, lacking synthetic controls, they rely solely on change in mass. In fact, these cross-links could occur anywhere and there is evidence to support that OxyC does indeed form the A-O-B and C-O-D cross-links. The evidence is circumstantial but has not been conclusively demonstrated.

Minor Issues

4. The abstract should be revised to be much more concise. Reading it on its own raised many questions that were subsequently addressed or clarified in the Introduction. For example, the sentence "Crosslinking of GPAs is typically performed by a cascade..." describes the "insertion of the 12 membered AB ring". This leads the viewer to the TOC image where unfortunately, the 12 membered ring is not present. Only later is the unique 15 membered ring of kistamicin addressed. Similarly, the the abstract distinguishes kistamicin from "medically active GPAs", which made me wonder about the differences and why kistamicin is being studied. I suggest shortening the Abstract, then including in the introduction some of the more pressing questions that justify study

- of the kistamicin pathway. These include the missing adenylation and epimerization domains (noted above) along with the existing discussion of the presence of only 2 rather than 3 P450s.
5. The chemdraw schematic of Figure S1 contains inconsistencies. Does module 6 insert a HPG (as shown in the assembly line) or a Tyr (as shown in the final kistamicin molecule? Similarly, the Trp residue is attached to C β through the C2 position in the assembly line and not C3 (as in the kistamicin).
 6. Figure 3. The D/E and F/G helices should be labeled within panel A.
 7. Figure 6. I believe the legend for panels C is incorrect. More broadly, it is not clear what the point of this section is. Wouldn't a single ligand demonstrate the formation of the thiolate at the enzyme active site?
 8. Figures 6 and S15. Are the binding constants for fluconazole and itraconazole well determined given the data in S15?
 9. The divergent activity of kistamicin is noted several places (both the introduction and the conclusions) but it is not clear whether this molecule has any known biological activity.
 10. Pg 34, section on extraction of kistamicin A. It is not clear to me what is intended by "cultivated 10 times in..."

Reviewer #2 (Remarks to the Author):

In the paper entitled 'Kistamicin biosynthesis reveals the biosynthetic requirements for production of highly active crosslinked glycopeptide antibiotics', Greule et. al. examine the oxidative cyclisation biosynthetic cascade in the glycopeptide kistamicin. Kistamicin differs from other related glycopeptides in that it possesses an ether linkage for the A-O-B ring as opposed to the typical biaryl coupling at this position. After sequencing the genome of the microbial producer, the authors identified two main discrepancies between the structure of the antibiotic and its corresponding proposed biosynthetic cluster. First they note a lacking epimerisation domain normally present and required in module four of the NRPS to epimerise and introduce a conserved hydroxyphenylglycine building block. The second, more substantial discrepancy revealed only two oxidative enzymes encoded in the cluster, suggesting that two of the three cyclisation reactions are likely catalysed by a single P450. Through heterologous expressions of an X-domain and the P450s OxyA and OxyC, the authors investigate the role of each oxidative enzyme through crystallographic analysis as well as by using model substrates to observe macrocyclisations. The authors solve a co-crystal structure of OxyA/X-domain and compare the interface with the respective interface observed in the complestatin system. The authors go on to claim OxyC is responsible for both the C-O-D and A-O-B ring formation from in vitro experiments, and delve further into substrate scope with comparisons with enzymes from chloroeremomycin biosynthesis. Lastly, the authors use molecular modeling of model peptides to suggest the larger A-O-B ring system in Type V glycopeptides is less stable than the A-B rings seen in Type I-VI systems.

While I appreciate the difficulty in working with these systems as well as the number of experiments reported in this manuscript, I do not recommend publication of this paper in Nature Communications. Most substantially, the authors do not provide appropriate evidence for the activity of OxyC and the corresponding enzymatic products to warrant the central claims of this paper that the enzyme is responsible for C-O-D and A-O-B ring formations. The bulk of the in vitro data for this enzyme is either poor or circumstantial, at best, and requires far more robust and in-depth characterisation to be acceptable for publication in my opinion (I provide specific examples below). More broadly, this paper is far too specialised for the audience of Nature Communications. This paper is an amalgamation of many different sets of (sometimes loosely related) experiments that end up distracting the reader. I feel this is at least two partially complete manuscripts rather than a single cohesive paper. It is my opinion that even if the data were more robust, this still would not be an appropriate paper for Nat. Comm.

Here are some of the major issues (in order of appearance); these comments do not touch upon more minor issues or the incongruences of all of the data presented:

1. The interaction assays that include Supplemental Figures S11-S13 either do not show

meaningful interactions/complex formation, or require additional analysis to support the claims. Fig. S11 shows a series of size exclusion runs of either single proteins or two proteins co-run. First, the authors should list the observed sizes/presumed oligomeric state of their proteins against a set of molecular weight standards. Even without this information, it is obvious that a complex of Oxy and the X-domain, which would elute earlier than either each domain separately, is not observed. The retention time of the X-domain remains essentially constant for all runs, and there is some apparent movement of OxyA elution to a size still smaller than the X-domain alone. Figure S12 is somewhat at odds with the size exclusion data (as mentioned in the manuscript) and might have some complexes observed that are pointed out with the arrows. But due to the banding of OxyA and OxyC, follow up verification of proteolytic digestion/LC-MS verification of the bands in question should be performed. In addition, to determine the sizes of the complex, the RF of the bands on multiple gels of different percentage acrylamide should be run with standards to determine the observed sizes. Figure S13 is also difficult to interpret, as there is little difference between the X-domain lanes from the lanes in combination with Oxy enzymes. Again, follow up MS analysis of the faint bands eluted to would ensure that both proteins were present at least, but this data shows a very minor fraction at best is crosslinked. Statements eluding to these experiments such as 'All three assays show clear evidence for an interaction between Oxykis enzymes and the Xkis-domain independent of the presence of the adjacent PCP domain' (Page 11), 'Closer analysis of the results of the analytical size exclusion experiments (which were supported by native PAGE) indicate a tighter interaction for OxyAkis with the kistamicin X-domain than OxyCkis: this can be seen in the elution of two distinct species in the case of OxyAkis (OxyAkis/Xkis in complex plus OxyA alone) whilst OxyCkis shows a broad peak of somewhat smaller elution volume than OxyCkis alone' (Page 11-12) and 'One possible explanation for the tight OxyAkis binding would be the nature of the aryl crosslink inserted in this case,...' are unfounded in my opinion.

2. The in vitro data reported in Figures 7 and S19 does not (yet) support the claims made by the authors that OxyC is responsible for C-O-D and A-O-B ring formations in kistamicin biosynthesis. The authors provide only low-resolution MS1 data for in vitro reactions on model tetrapeptide phenyl/aryl substrates. The authors report losses of 2 and/or 4 Da and assume (1) substrate specificity for D- over L-Hpg is observed in these experiments and (2) these masses refer to phenyl ether bond formation for the expected model C-O-D and A-O-B bonds. In order to state these reactions are producing the products they claim, they require, at the very least, discernable high-resolution MSMS confirmation, if not NMR data to support their claims. As they synthesized tetrapeptides in these experiments, MSMS fragmentation of the bicycled product would not work, if in fact they were modified as assumed. Post-reaction derivatization with phenyl-reactive agents such as chloroformates would be required to try and tease out the singly modified peptide orientations. Notwithstanding this requirement, I do not believe the data supports selectivity of D-Hpg over L-Hpg at this time. The authors do not have normalized X-axes in Figure S19, and if they were, the relative peak heights for peaks corresponding to the monocyclised masses would be roughly equivalent for D-tetrapeptide 3 and L-tetrapeptide 4 in their OxyC reactions. It is difficult to determine the relative loss of substrate in the reactions due to substantial peak height/area differences (which call into question the relative loading onto the PCP/derivatization/reisolation of the substrates in these reactions). The authors also do not address observing multiple (3 or 4) peaks for the presumed monocyclised species. The authors only point to a single species for the presumed bicycled product in figures 7 and S19, but could easily point out other peaks in these spectra due to the noise in the data. The data presented in Figure 8 and Figure S20 has similar failings as pointed out for Figures 7 and S19. The substrate includes at least one residue with the incorrect stereochemistry and without chlorination (the timing of which was shown critical in vancomycin biosynthesis in reference 17 of this manuscript). Here the authors test OxyBtei from teicoplanin biosynthesis and compares the activity with OxyCkis. The authors do not explain why a single peak is observed with OxyBtei and at least three peaks are pointed out in the OxyC reactions. The peak shifts of the linear masses in the OxyBtei reactions are also peculiar and not addressed. Again, these data require high-res MSMS analysis or NMR structural information to validate the authors' claims.

3. For the data presented in Figures 9 and S21, the authors try and tease out substrate scope of OxyC with even more non-natural substrates. The ability to tease out any sort of substrate scope to make comparisons to chloroeremomycin biosynthetic enzymes simply cannot be made with the above-mentioned lack of characterisation of the reaction products.

Reviewer #3 (Remarks to the Author):

This is an impressive manuscript that explores unusual peptide crosslink biochemistry in the antibiotic kistamicin, a member of the larger glycopeptide family. The authors observe that while kistamicin has three crosslinks, it differs from other members of the family in that it only encodes two candidate cross linking P450 enzymes in the biosynthetic gene cluster. The Cryle lab has pioneered the study of the glycopeptide biosynthesis P450s and revealed among other things, the importance of the accessory X-domain in cognate cross linking reactions . In this work they dissect the roles of the two P450s biochemically and show that one enzyme, OxyC, performs an unprecedented two cross linking reactions, one a C-C link and the other a phenolic O-C link. The authors very nicely combine genomic, biochemical, and protein structural strategies to address this fascinating and new aspect of glycopeptide antibiotic chemistry. This is a worthy addition too our understanding of how this important antibiotics are produced.

Come comments for consideration:

Line 259-260: In lines 295-297, the authors suggest that a peptide with the correct crosslinked state is required for tight binding between the X domain and correct Oxy enzyme. Since there were no peptide structures in the gel filtration/native PAGE binding experiments, could this affect the relative binding affinities of OxyA and OxyC so that they are not necessarily reflective of physiological affinities with peptide present?

Line 421-428 and Figure 6: Since this experiment isn't very informative, consider moving to supplemental info.

Figure 7, Line 478: Why are there two peaks that correspond to the monocyclic product? Are they the two different crosslinks that could be formed by OxyC?

Figure 8B: Why are there three peaks with different retention times that represent the monocyclic product? Why is there a shift in retention time of the linear and monocyclic products in the top row versus the bottom row?

Line 568-573 and Figure 9B: Could the inability of OxyCkis to install the A-O-B ring in this system be do to either reaction conditions? Even in Figure 8 with a OxyCkis more natural substrate, it works with very low efficiency. Could it also have to do with the order of ring formation? The authors used OxyB to introduce the C-O-D ring, but maybe OxyCkis works better by first installing an A-O-B link before the C-O-D link. Changing the order of incubating the different enzymes in this system and with a natural substrate control might be able to tease this apart.

Minor issues:

Line 120-121 and Line 544: "Limited antibiotic activity" is vague. Originally reported MICs against *S. aureus* for complestatin is ~2ug/mL, and kistamicin is 12.5-25ug/mL

Line 141-142: The authors imply that the AB ring versus the A-O-B is essential for GPA's antibiotic activity. It being essential isn't necessarily true, since kistamicin and complestatin still have antibiotic activity.

Line 178: Specify OxyAkis to avoid confusion with referring to a family of enzymes

Line 558, 569: Which GPA's OxyA and OxyB enzymes are used?

Response to Reviewers' comments:

We would like to thank both the editorial staff and the reviewers for their time and critical insights into our manuscript. In this revision, we have been able to significantly improve the manuscript through the inclusion of (1) *in vivo* gene disruption/ complementation experiments to probe the cyclisation of the kistamicin peptide during biosynthesis complete with high resolution mass spectrometry (HRMS) and MS/MS fragmentation analysis to confirm the intermediates identified, (2) extended *in vitro* characterisation of the activity of the OxyC_{kis} enzyme, with the structures of key products also confirmed by HRMS and MS/MS fragmentation analysis, and (3) comprehensive analysis of Oxy/X-domain interactions through the use of isothermal titration calorimetry (ITC) and fluorescently protein labelling in native PAGE assays. The manuscript has been completely re-written to distinguish results from discussion and to assist in maintaining the narrative of the paper. We believe that this revised manuscript is substantially improved and addresses the all of the experimental concerns raised by the reviewers within their original reviews.

Reviewer #1 (Remarks to the Author):

The manuscript by Cryle and colleagues continues their structural and functional investigation of the P450 enzymes that are involved in cross-link formation between aromatic residues of the glycopeptide antibiotics. Previously, the authors have studied primarily teicoplanin biosynthesis and determined the order of cross-link formation, the structures of several enzymes involved and the rather interesting presence of a non-functional condensation domain variant called the X-domain that serves as an protein-protein interface to capture the P450s and direct them towards the substrate.

The current paper continues this work in a different system that is responsible for the production of kistamicin. There are several unusual features of this NRPS pathway, some of which are experimentally explored herein. The authors present a crystal structure of the OxyA P450 bound to the X-domain. This is the first OxyA type protein (unlike the OxyB of the teichoplanin in earlier work) crystallized bound to the X-domain. This confirms a similar interaction interface suggesting only one P450 can bind the X-domain at a time. Functional analysis then investigates an outstanding question in kistamicin biosynthesis, namely the presence of three cross-links but only two P450 proteins.

The paper requires significant revision as noted below. The structural determination and analysis is sound. However some of the introductory material is speculative and, as currently written, runs the risk of becoming accepted as proven without experimental evidence. Further, some of the protein interaction and functional analysis does not fully support the conclusions drawn. Finally, the paper would benefit from significant revision to improve clarity. These issues are described below.

Significant Issues

1. The presentation of the potential activation of the the Tyrosine residue by a Tyr-RS is speculative but presented as conclusive, as is its epimerization by the condensation domain.

Further, the lack of an E-domain in module 4 is rationalized with a dual-function C/E domain downstream. These are interesting variations but should not be presented in “Results” until they are experimentally addressed.

The results and discussion has been now divided to provide better clarity around this and related issues. This was a helpful point and we thank the reviewer for this suggestion, which aids in improving the clarity of the revised manuscript.

2. The authors examine a functional interaction between OxyA and OxyC P450s and various protein constructs. The gel shifts in Figure S12 do not support the conclusion, particularly with OxyA, of a functional interaction. The shifted band (arrow in OxyA + PCP-X) matches the band in the OxyA lane alone. This analysis is challenged by the ladder of bands from oligomers. Perhaps if this experiment had been run with much less X or PCP-X, the disappearance of the band for the unshifted protein would be more convincing. Similarly, the long arrows in Figure S13 (right panel) for the chemical cross-linking are likely to point to large aggregates of protein that failed to enter the running gel.

Given the challenges of the interpretation of the original experiments, we undertook several modified as well as new experiments to quantify the interaction of these proteins. Firstly, we removed the fusion MBP tags and utilised a fluorescein labelling strategy for the NRPS protein components to allow better resolution of the components in both the native PAGE as well as analytical size exclusion experiments. These (in particular in the case of the native-PAGE experiments) greatly assist in data interpretation. Furthermore, we initiated a collaboration to examine the X/Oxy interface using isothermal titration calorimetry (ITC), and have performed this for both the pairings of OxyA+X and OxyC+X. The data from these experiments supports the interaction of the Oxy enzymes with the X-domain, and indicate a 1:1 binding interaction with a K_d in the micromolar range – this is in line with other measurements of Oxy/X-domain affinity from the teicoplanin system using complementary techniques. These results also support the interface seen in the novel OxyA/X crystal structure presented within this work.

3. The functional analyses of Figures 7 and 8 are challenging. The authors attempt to demonstrate cross-linking being catalyzed by the OxyC protein. They conclude that they have shown this however, lacking synthetic controls, they rely solely on change in mass. In fact, these cross-links could occur anywhere and there is evidence to support that OxyC does indeed form the A-O-B and C-O-D cross-links. The evidence is circumstantial but has not been conclusively demonstrated.

We have undertaken several new experiments to better resolve the activity of the OxyC_{kis} enzyme, including *in vivo* gene disruption/ complementation experiments and further *in vitro* turnover assays with new substrates. In the gene disruption experiments, we first generated two mutant strains in which either the gene encoding OxyA (*kisN*) or OxyC (*kisO*) was deleted, with these strains then cultivated and the products extracted before being analysed by high resolution MS and MS/MS. In both strains, the production of kistamicin was abolished. In the $\Delta kisO$ strain only linear peptide intermediates were detected, and hence we

concentrated on the $\Delta kisN$ strain. In the $\Delta kisN$ strain monocyclic intermediates could be detected, whilst the strain complemented with a plasmid expressing *kisN* restores kistamicin production. MS/MS fragmentation and analysis showed that for both hexa and heptapeptide intermediates the crosslink present was the C-O-D ring installed between Tyr-6 and Hpg-4. This indicates that – in agreement with *in vivo* gene deletion experiments performed for other GPA producers – the first ring inserted in the peptide is the C-O-D ring and that in the kistamicin system this is performed by OxyC_{kis}. Low levels of a bicyclic product could be detected from the $\Delta kisN$ strain, which indicates the ability of the OxyC enzyme to install multiple rings within one peptide. However, as this is a hexapeptide intermediate this product contains an alternate ring to the A-O-B ring present in kistamicin. We interpret this as being due to the order of reaction of crosslinking matching that of other GPAs, i.e. (1) C-O-D, (2) DE, (3) A-O-B. We further investigated this crosslinking activity *in vitro* using new peptide substrates and HRMS + MS/MS analysis of key products. This confirmed that OxyC_{kis} is highly effective at cyclising linear peptides and yet appears to require the DE ring prior to any substantial bicyclisation activity. We do note, however, that OxyC_{kis} is highly effective at cyclising a variety of peptides (even non-standard ones) and appears highly promiscuous in its activity. We also confirmed that OxyC_{kis} could indeed act on the N-terminal portion of the kistamicin peptide, which also indicates that the rigidity of the cyclised peptide helps to constrain the activity of the OxyC_{kis} enzyme towards one product *in vivo*. We then undertook further complementation experiments to examine the specificity of Type I-IV Oxy enzymes in comparison to those from the kistamicin producer. Further complementation of the wildtype producer with a plasmid containing the *staG* (*oxyE*) gene did not lead to the insertion of the F-O-G ring, and the kistamicin Oxy enzymes could not restore antibiotic production in related OxyA/OxyC deletion strains of balhimycin and A47934. These experiments indicate that these enzymes are highly specific to the structure of the partially crosslinked peptides and also that the promiscuity of the kistamicin OxyC enzyme is held in check in the natural biosynthetic pathway by the increasing rigidity of the substrate during the cyclisation pathway. We have altered the discussion to indicate that these results are suggestive for OxyC_{kis} activity in forming the A-O-B ring but that currently this activity is unable to be directly assayed due to experimental challenges in obtaining active OxyA_{kis} enzyme or the highly complex bicyclic peptide intermediate substrate required here.

Minor Issues

4. *The abstract should be revised to be much more concise. Reading it on its own raised many questions that were subsequently addressed or clarified in the Introduction. For example, the sentence “Crosslinking of GPAs is typically performed by a cascade...” describes the “insertion of the 12 membered AB ring”. This leads the viewer to the TOC image where unfortunately, the 12 membered ring is not present. Only later is the unique 15 membered ring of kistamicin addressed. Similarly, the the abstract distinguishes kistamicin from “medically active GPAs”, which made me wonder about the differences and why kistamicin is being studied. I suggest shortening the Abstract, then including in the introduction some of the more pressing questions that justify study of the kistamicin pathway.*

These include the missing adenylation and epimerization domains (noted above) along with the existing discussion of the presence of only 2 rather than 3 P450s.

We have shortened the abstract and also adjusted the introduction as suggested.

5. The chemdraw schematic of Figure S1 contains inconsistencies. Does module 6 insert a HPG (as shown in the assembly line) or a Tyr (as shown in the final kistamicin molecule? Similarly, the Trp residue is attached to C β through the C2 position in the assembly line and not C3 (as in the kistamicin).

These errors have been corrected.

6. Figure 3. The D/E and F/G helices should be labeled within panel A.

These have now been labelled in the revised figure.

7. Figure 6. I believe the legend for panels C is incorrect. More broadly, it is not clear what the point of this section is. Wouldn't a single ligand demonstrate the formation of the thiolate at the enzyme active site?

We have now removed this figure from the main text, with the inhibitor binding data for the kistamicin Oxy enzymes now found in the SI.

8. Figures 6 and S15. Are the binding constants for fluconazole and itraconazole well determined given the data in S15?

Given the low absorption change on the binding of these two figures, we have removed the binding constants shown in the SI for these inhibitors and have indicated that the absorption change is too low in these cases to allow such calculations to be reliable.

9. The divergent activity of kistamicin is noted several places (both the introduction and the conclusions) but it is not clear whether this molecule has any known biological activity.

We now indicate the activities reported for both Type V GPAs kistamicin and complestatin in the text, which share antibacterial activity as well as antiviral activity. This suggests that there is definitely scope for GPAs with altered structures and crosslinking to have different biological activity, which shows the importance of characterising these biosynthetic machineries and we thank the reviewer for making this point.

10. Pg 34, section on extraction of kistamicin A. It is not clear to me what is intended by "cultivated 10 times in..."

This error has been corrected.

Reviewer #2 (Remarks to the Author):

In the paper entitled 'Kistamicin biosynthesis reveals the biosynthetic requirements for production of highly active crosslinked glycopeptide antibiotics', Greule et. al. examine the oxidative cyclisation biosynthetic cascade in the glycopeptide kistamicin. Kistamicin differs from other related glycopeptides in that it possesses an ether linkage for the A-O-B ring as opposed to the typical biaryl coupling at this position. After sequencing the genome of the microbial producer, the authors identified two main discrepancies between the structure of the antibiotic and its corresponding proposed biosynthetic cluster. First they note a lacking epimerisation domain normally present and required in module four of the NRPS to epimerise and introduce a conserved hydroxyphenylglycine building block. The second, more substantial discrepancy revealed only two oxidative enzymes encoded in the cluster, suggesting that two of the three cyclisation reactions are likely catalysed by a single P450. Through heterologous expressions of an X-domain and the P450s OxyA and OxyC, the authors investigate the role of each oxidative enzyme through crystallographic analysis as well as by using model substrates to observe macrocyclisations. The authors solve a co-crystal structure of OxyA/X-domain and compare the interface with the respective interface observed in the complestatin system. The authors go on to claim OxyC is responsible for both the C-O-D and A-O-B ring formation from in vitro experiments, and delve further into substrate scope with comparisons with enzymes from chloroeremomycin biosynthesis. Lastly, the authors use molecular modeling of model peptides to suggest the larger A-O-B ring system in Type V glycopeptides is less stable than the A-B rings seen in Type I-VI systems.

While I appreciate the difficulty in working with these systems as well as the number of experiments reported in this manuscript, I do not recommend publication of this paper in Nature Communications. Most substantially, the authors do not provide appropriate evidence for the activity of OxyC and the corresponding enzymatic products to warrant the central claims of this paper that the enzyme is responsible for C-O-D and A-O-B ring formations. The bulk of the in vitro data for this enzyme is either poor or circumstantial, at best, and requires far more robust and in-depth characterisation to be acceptable for publication in my opinion (I provide specific examples below). More broadly, this paper is far too specialised for the audience of Nature Communications. This paper is an amalgamation of many different sets of (sometimes loosely related) experiments that end up distracting the reader. I feel this is at least two partially complete manuscripts rather than a single cohesive paper. It is my opinion that even if the data were more robust, this still would not be an appropriate paper for Nat. Comm.

Here are some of the major issues (in order of appearance); these comments do not touch upon more minor issues or the incongruences of all of the data presented:

1. The interaction assays that include Supplemental Figures S11-S13 either do not show meaningful interactions/complex formation, or require additional analysis to support the claims. Fig. S11 shows a series of size exclusion runs of either single proteins or two proteins co-run. First, the authors should list the observed sizes/presumed oligomeric state of their proteins against a set of molecular weight standards. Even without this information, it is obvious that a complex of Oxy and the X-domain, which would elute earlier than either each

domain separately, is not observed. The retention time of the X-domain remains essentially constant for all runs, and there is some apparent movement of OxyA elution to a size still smaller than the X-domain alone. Figure S12 is somewhat at odds with the size exclusion data (as mentioned in the manuscript) and might have some complexes observed that are pointed out with the arrows. But due to the banding of OxyA and OxyC, follow up verification of proteolytic digestion/LC-MS verification of the bands in question should be performed. In addition, to determine the sizes of the complex, the RF of the bands on multiple gels of different percentage acrylamide should be run with standards to determine the observed sizes. Figure S13 is also difficult to interpret, as there is little difference between the X-domain lanes from the lanes in combination with Oxy enzymes. Again, follow up MS analysis of the faint bands eluted to would ensure that both proteins were present at least, but this data shows a very minor fraction at best is crosslinked. Statements eluding to these experiments such as 'All three assays show clear evidence for an interaction between Oxykis enzymes and the Xkis-domain independent of the presence of the adjacent PCP domain' (Page 11), 'Closer analysis of the results of the analytical size exclusion experiments (which were supported by native PAGE) indicate a tighter interaction for OxyA_{kis} with the kistamicin X-domain than OxyC_{kis}: this can be seen in the elution of two distinct species in the case of OxyA_{kis} (OxyA_{kis}/Xkis in complex plus OxyA alone) whilst OxyC_{kis} shows a broad peak of somewhat smaller elution volume than OxyC_{kis} alone' (Page 11-12) and 'One possible explanation for the tight OxyA_{kis} binding would be the nature of the aryl crosslink inserted in this case, ...' are unfounded in my opinion.

We agree that the information contained in these original studies was limited. To help address this, we have now performed ITC experiments that confirm X-domain binding to OxyC_{kis} and OxyA_{kis} in the low micromolar range. We have also adopted an X-domain labelling strategy that allows improved analysis of both the gel filtration and native PAGE experiments. These experiments thus greatly improve this section of the manuscript. Protein standards run on SDS-PAGE have also now been included in the manuscript as suggested.

2. The *in vitro* data reported in Figures 7 and S19 does not (yet) support the claims made by the authors that OxyC is responsible for C-O-D and A-O-B ring formations in kistamicin biosynthesis. The authors provide only low-resolution MS1 data for *in vitro* reactions on model tetrapeptide phenyl/aryl substrates. The authors report losses of 2 and/or 4 Da and assume (1) substrate specificity for D- over L-Hpg is observed in these experiments and (2) these masses refer to phenyl ether bond formation for the expected model C-O-D and A-O-B bonds. In order to state these reactions are producing the products they claim, they require, at the very least, discernable high-resolution MSMS confirmation, if not NMR data to support their claims. As they synthesized tetrapeptides in these experiments, MSMS fragmentation of the bicycled product would not work, if in fact they were modified as assumed. Post-reaction derivatization with phenyl-reactive agents such as chloroformates would be required to try and tease out the singly modified peptide orientations. Notwithstanding this requirement, I do not believe the data supports selectivity of D-Hpg over L-Hpg at this time. The authors do not have normalized X-axes in Figure S19, and if they were, the relative peak heights for peaks corresponding to the monocyclised masses would be roughly equivalent for

D-tetrapeptide 3 and L-tetrapeptide 4 in their OxyC reactions. It is difficult to determine the relative loss of substrate in the reactions due to substantial peak height/area differences (which call into question the relative loading onto the PCP/derivatization/reisolation of the substrates in these reactions). The authors also do not address observing multiple (3 or 4) peaks for the presumed monocyclised species. The authors only point to a single species for the presumed bicyclised product in figures 7 and S19, but could easily point out other peaks in these spectra due to the noise in the data. The data presented in Figure 8 and Figure S20 has similar failings as pointed out for Figures 7 and S19. The substrate includes at least one residue with the incorrect stereochemistry and without chlorination (the timing of which was shown critical in vancomycin biosynthesis in reference 17 of this manuscript). Here the authors test OxyBtei from teicoplanin biosynthesis and compares the activity with OxyCkis. The authors do not explain why a single peak is observed with OxyBtei and at least three peaks are pointed out in the OxyC reactions. The peak shifts of the linear masses in the OxyBtei reactions are also peculiar and not addressed. Again, these data require high-res MSMS analysis or NMR structural information to validate the authors' claims.

To clarify the identities of key peptide products we have performed HRMS and MS/MS analyses to support the structures of key intermediates; these experiments clearly show that OxyC_{kis} installs the initial C-O-D ring into the linear peptide precursor. We have demonstrated the effect of the Oxy deletions *in vivo*, and also shown that the OxyC_{kis} enzyme can be promiscuous both *in vivo* and *in vitro*. The lack of high levels of bicyclisation combined with new *in vivo* and *in vitro* data still support the role of OxyC in the bicyclisation process, and also indicate that there is a restriction on the second bicyclisation step that requires the presence of the DE ring first. This is consistent with other reported *in vivo* GPA gene disruption experiments.

3. For the data presented in Figures 9 and S21, the authors try and tease out substrate scope of OxyC with even more non-natural substrates. The ability to tease out any sort of substrate scope to make comparisons to chloroeremomycin biosynthetic enzymes simply cannot be made with the above-mentioned lack of characterisation of the reaction products.

Given the significant amount of new data now included in the revised manuscript as well as the initial opinion of the reviewer that the narrative was somewhat lacking in our submitted manuscript, we have now downplayed the results concerning the comparative activities of the different OxyC homologues to maintain the clarity and flow of the manuscript. We have also added new peptide turnovers for the OxyC_{kis} enzyme that support the activity of this enzyme seen *in vivo*, with important major peptide products now verified by HRMS and MS/MS analyses.

Reviewer #3 (Remarks to the Author):

This is an impressive manuscript that explores unusual peptide crosslink biochemistry in the antibiotic kistamicin, a member of the larger glycopeptide family. The authors observe that while kistamicin has three crosslinks, it differs from other members of the family in that it

only encodes two candidate cross linking P450 enzymes in the biosynthetic gene cluster. The Cryle lab has pioneered the study of the glycopeptide biosynthesis P450s and revealed among other things, the importance of the accessory X-domain in cognate cross linking reactions. In this work they dissect the roles of the two P450s biochemically and show that one enzyme, OxyC, performs an unprecedented two cross linking reactions, one a C-C link and the other a phenolic O-C link. The authors very nicely combine genomic, biochemical, and protein structural strategies to address this fascinating and new aspect of glycopeptide antibiotic chemistry. This is a worthy addition too our understanding of how this important antibiotics are produced.

Some comments for consideration:

Line 259-260: In lines 295-297, the authors suggest that a peptide with the correct crosslinked state is required for tight binding between the X domain and correct Oxy enzyme. Since there were no peptide structures in the gel filtration/native PAGE binding experiments, could this affect the relative binding affinities of OxyA and OxyC so that they are not necessarily reflective of physiological affinities with peptide present?

We have now performed ITC analyses and added this data to the revised version, which indicates a low micromolar affinity for both P450s to the X-domain that support the measurements we have made with the teicoplanin system using other assays. We have previously shown that Oxy enzymes compete for X-domain binding, and that the peptide substrate is a crucial factor in mediating interactions with the correct Oxy enzyme, and the data we have obtained here supports this model. Although it is important to note that the peptide association with the Oxy enzyme is PCP-mediated, not directly through the X-domain.

Line 421-428 and Figure 6: Since this experiment isn't very informative, consider moving to supplemental info.

This figure has been moved to the supplemental information and the results/ discussion about Oxy/ inhibitor binding greatly reduced in the revised manuscript.

Figure 7, Line 478: Why are there two peaks that correspond to the monocyclic product? Are they the two different crosslinks that could be formed by OxyC?

HRMS and MS/MS analyses have shown that different products can be produced depending on the peptide used in the *in vitro* turnovers with OxyC_{kis}, along with products that have formed due to the epimerisation of Hpg-residues within the peptide substrates. From the data that is included in this revised manuscript, OxyC_{kis} appears to be highly promiscuous in forming different crosslinks within relatively small peptides, which stands in contrast to the OxyB/C enzymes from Type I-IV GPAs.

Figure 8B: Why are there three peaks with different retention times that represent the

monocyclic product? Why is there a shift in retention time of the linear and monocyclic products in the top row versus the bottom row?

We have repeated all the turnover experiments now in the same batch run and with the same conditions to minimise differences between experiments. The presence of multiple peaks within the K4-7 tetrapeptide and K1-6 hexapeptide all show the same fragmentation in MS/MS and indicate the presence of the C-O-D ring, from which we deduce that the multiple/ broad peaks in these cases are the result of Hpg/Dpg epimerisation due to the high number of these residues in these peptides and their relative ease of racemisation. In the case of the K1-4 tetrapeptide, these multiple peaks were shown to be the result of 1-3 and 1-4 crosslinks as well as Hpg/Dpg epimerisation, which shows that OxyC_{kis} enzyme is a promiscuous crosslinking enzyme.

Line 568-573 and Figure 9B: Could the inability of OxyC_{kis} to install the A-O-B ring in this system be do to either reaction conditions? Even in Figure 8 with a OxyC_{kis} more natural substrate, it works with very low efficiency. Could it also have to do with the order of ring formation? The authors used OxyB to introduce the C-O-D ring, but maybe OxyC_{kis} works better by first installing an A-O-B link before the C-O-D link. Changing the order of incubating the different enzymes in this system and with a natural substrate control might be able to tease this apart.

We agree that the lack of A-O-B activity from OxyC_{kis} *in vitro* likely stems from the order of reactions that is adopted *in vivo*. Our gene disruption experiments show that that the OxyA_{kis} enzyme acts after the initial OxyC_{kis} catalysed C-O-D ring formation, with comparison with the complestatin system supporting this activity as DE ring formation. This order of activity means that *in vivo* there is no direct way to assay OxyC_{kis} A-O-B ring formation as both enzymes are required to produce the bicyclic intermediate required. Furthermore, activity of the OxyA_{kis} enzyme is not able to be reconstituted *in vitro*, which limits our ability to probe this step. One possible option would be to introduce other Oxy enzymes into the kistamicin deletion strains to examine bicyclisation, however these experiments have proven to be challenging thus far, and access to a synthetic bicyclic intermediate is also a highly challenging synthesis in its own right. We believe that the data we have added here strongly supports OxyC_{kis} as the A-O-B cyclising enzyme, but stress that this evidence – whilst strong – is not a direct demonstration of this cyclisation activity.

Minor issues:

Line 120-121 and Line 544: "Limited antibiotic activity" is vague. Originally reported MICs against S. aureus for complestatin is ~2ug/mL, and kistamicin is 12.5-25ug/mL

We thank the reviewer for this point and have now listed the activity for these Type V GPAs along with an improved explanation of their activities, including antiviral effects.

Line 141-142: The authors imply that the AB ring versus the A-O-B is essential for GPA's

antibiotic activity. It being essential isn't necessarily true, since kistamicin and complestatin still have antibiotic activity.

We have clarified this now to indicate that the AB ring is essential for the activity of Type I-IV GPAs through inhibition of bacterial cell wall biosynthesis mediated through binding to lipid II. We now also include a reference to the fact that the antibiotic activity of complestatin is mediated through inhibition of the fatty acid biosynthesis pathway, although the mechanism of antibiotic activity displayed by kistamicin is as of yet unclear.

Line 178: Specify OxyA_{kis} to avoid confusion with referring to a family of enzymes

We have now specific the enzyme to minimise possible confusion.

Line 558, 569: Which GPA's OxyA and OxyB enzymes are used?

The Oxy enzyme used were from teicoplanin biosynthesis

Reviewers' comments:

Reviewer #1 (Remarks to the Author):

The authors have addressed all of my prior concerns. The revised manuscript more clearly separates the speculative information into the Discussion section, resulting in an improved manuscript that distinguishes the direct conclusions from other interesting aspects of kistamycin biosynthesis that remain to be elucidated.

The authors should note that the chemical structures representing Tryptophan side chains are still incorrect, I believe, in

- Figure 9, the left and middle peptide structures
- Figure S1, modules 2-7

Reviewer #2 (Remarks to the Author):

I concur with the authors that the resubmitted manuscript is a serious improvement from the original submission. The inclusion of the added experiments, specifically the knockouts, fluorescence tagging, ITC, MS/MS analysis, and more extensive in vitro assays has significantly strengthened the paper. In addition, the rewriting of the manuscript – and the appropriately tempered tone of their interpretation of the results – is to be commended.

I do have several comments/suggestions for the added experiments:

1) Figure 5 shows both the improved native-PAGE experiments with fluorescently tagged X and PCP-X domains as well as the ITC experiments with the X domain. The authors state in the main text: "These assays showed evidence of an interaction between the kistamicin Oxy enzymes and the X-domain independent of the presence of the adjacent PCP domain. This was evidenced by the formation of bands of higher molecular weight in the lanes where X-domain and Oxy enzymes were present (Figure 5A-C)." However, from my interpretation, the data in Figure 5A-C clearly show substantially stronger OxyA/C associations with the PCP-X domain than the X-domain alone. The authors, at the very least, should directly address this in the main text. And based on this fact, I would expect to see ITC data for OxyA/C associations with the PCP-X domain so readers can appropriately compare the sets of experiments.

2) The figure legends for Figures S17, S20, S22, and S24 need to more explicitly describe the respective figures. For instance, the authors should point out the respective loss or gain of peaks in the upper-right panels compared to the upper left panels (and mention the differently scaled y-axes) and what those results mean. Also, the X-axes are not all uniform in size as they should be. For the bottom left panels, the authors should describe what the 'Expected' column is denoting – it appears to be a theoretical isotopic distribution, but why/how that relates to the intensity is not clear.

3) For Figures S21 and S23, the chemical structures for the fragmented peptides with the various fragmentations should be added to these figures for clarity. Also, the authors should note in the figure legend of S23 the 'x5' and 'x50' sections of the spectrum. If these peaks are magnified in these mass ranges as I suspect, those mass ranges should be in in-set windows as to not look as if the spectrum is continuous.

4) The authors should include MS/MS data for the bicyclic hexapeptide in their delta-kisN mutant. The ion count as seen in Figure S25 should be sufficient for fragmentation, and fragmentation between residues 3 and 4 of the hexapeptide would provide further evidence for the cyclization pattern and structure drawn in Figure 3.

5) Lastly, Figure S31 and Table S5 describe the size-exclusion chromatography experiments with the different interacting domains (with the MBP tags now cleaved off). From what I can tell from these graphs, these experiments show essentially no discernable/considerable interaction or appropriately sized complexes and should be explicitly pointed out in the main text. I am at a loss how this data is to be interpreted, especially without the relative sizes of the complexes noted based on a standard curve of molecular weight standards.

Reviewer #3 (Remarks to the Author):

The authors have adequately addressed the main points in my initial review. The only additional note I should make is that the fatty acid inhibition data in ref 25 newly added in this revision is questionable. It is perhaps unfair to point this out as this reference is peer reviewed, but I fear to cite it amplifies a message that is ambiguous.

Response to Reviewers' comments:

We would once again like to thank the editorial staff and the reviewers for their time and analysis of our revised manuscript. We were pleased to note that the improvements we had incorporated in the original revision to this manuscript were able to address the concerns of all reviewers, and that in this revision we have now been able to address all the remaining points that remain central to the thesis of this work.

Reviewer #1 (Remarks to the Author):

The authors have addressed all of my prior concerns. The revised manuscript more clearly separates the speculative information into the Discussion section, resulting in an improved manuscript that distinguishes the direct conclusions from other interesting aspects of kistamycin biosynthesis that remain to be elucidated.

We are very pleased that we could address the previous concerns of this reviewer. Are grateful for their suggestions that have significantly improved the manuscript.

The authors should note that the chemical structures representing Tryptophan side chains are still incorrect, I believe, in

- *Figure 9, the left and middle peptide structures*
- *Figure S1, modules 2-7*

Thank you for identifying this mistake – all instances have now been corrected.

Reviewer #2 (Remarks to the Author):

I concur with the authors that the resubmitted manuscript is a serious improvement from the original submission. The inclusion of the added experiments, specifically the knockouts, fluorescence tagging, ITC, MS/MS analysis, and more extensive in vitro assays has significantly strengthened the paper. In addition, the rewriting of the manuscript – and the appropriately tempered tone of their interpretation of the results – is to be commended.

We again are very pleased to have been able to improve the manuscript in line with the comments of this reviewer.

I do have several comments/suggestions for the added experiments:

1) Figure 5 shows both the improved native-PAGE experiments with fluorescently tagged X and PCP-X domains as well as the ITC experiments with the X domain. The authors state in the main text: “These assays showed evidence of an interaction between the kistamicin Oxy enzymes and the X-domain independent of the presence of the adjacent PCP domain. This was evidenced by the formation of bands of higher molecular weight in the lanes where X-domain and Oxy enzymes were present (Figure 5A-C).” However, from my interpretation, the data in Figure 5A-C clearly show substantially stronger OxyA/C associations with the PCP-X domain than the X-domain alone. The authors, at the very least, should directly address this in the main text. And based on this fact, I would expect to see ITC data for OxyA/C

associations with the PCP-X domain so readers can appropriately compare the sets of experiments.

We are happy to see that the reviewer agrees that the native-PAGE experiments clearly support interactions between the Oxy enzymes and constructs containing the X-domain. Our AUC data also supports that there is an interaction between the Oxy enzymes and the X-domain. Beyond this, we have not attempted to quantify the native-PAGE data as we believe such interpretation would be prone to significant error. The purpose of including these experiments in the manuscript was to obtain evidence supporting the relevance of the OxyA/X-domain complex structure reported in this paper. It has clearly done this and further AUC experiments to determine PCP-X/ Oxy interactions would rather be probing the quantitative effect of the PCP on this interaction, which is not central to the discussion and significant quantity of results already contained in this manuscript.

2) The figure legends for Figures S17, S20, S22, and S24 need to more explicitly describe the respective figures. For instance, the authors should point out the respective loss or gain of peaks in the upper-right panels compared to the upper left panels (and mention the differently scaled y-axes) and what those results mean. Also, the X-axes are not all uniform in size as they should be. For the bottom left panels, the authors should describe what the 'Expected' column is denoting – it appears to be a theoretical isotopic distribution, but why/how that relates to the intensity is not clear.

We have altered these figure legends in line with the comments of this reviewer. Furthermore, we have also included a section of text (SI Page 23) by way of introduction to these data in the supplement to clarify and explain the data that is contained in each of these figures.

3) For Figures S21 and S23, the chemical structures for the fragmented peptides with the various fragmentations should be added to these figures for clarity. Also, the authors should note in the figure legend of S23 the 'x5' and 'x50' sections of the spectrum. If these peaks are magnified in these mass ranges as I suspect, those mass ranges should be in in-set windows as to not look as if the spectrum is continuous.

We have included key ion structures for figures S21 and S23 as requested, with the remaining fragments common ions in mass spectrometry. Thus, we have not annotated every peak beyond including the names of each fragment, and have added two references explaining this nomenclature in the new MS introduction section now included in the supplement. We have also included a new version of figure S23 with the magnification removed and insets highlighting the key ions.

4) The authors should include MS/MS data for the bicyclic hexapeptide in their delta-kisN mutant. The ion count as seen in Figure S25 should be sufficient for fragmentation, and fragmentation between residues 3 and 4 of the hexapeptide would provide further evidence for the cyclization pattern and structure drawn in Figure 3.

We have included this data in the supplement as requested. In addition (and to avoid potential confusion with the interpretation of Figure 3), we now indicate that several bicyclic peptides have been identified from *in vivo* experiments (which was previously only mentioned in the text) and show the second ring formed as dotted/dashed lines to indicate this.

5) Lastly, Figure S31 and Table S5 describe the size-exclusion chromatography experiments with the different interacting domains (with the MBP tags now cleaved off). From what I can tell from these graphs, these experiments show essentially no discernable/considerable interaction or appropriately sized complexes and should be explicitly pointed out in the main text. I am at a loss how this data is to be interpreted, especially without the relative sizes of the complexes noted based on a standard curve of molecular weight standards.

We agree that these data are not easily interpreted without considerable further effort, although they in no way conflict with our interaction hypothesis. In addition, the limited information content of these experiments is now essentially redundant when compared to that provided by the new labelled native-PAGE and Oxy/X AUC data now included in the manuscript (Figure 5) in response to referee comments. Given this, we have simply removed the size-exclusion experiments from the supplementary data. With significant new data incorporated into the supplement, we also have removed theazole binding figures previously included in the supplement as these are outside of the scope of this revised manuscript.

Reviewer #3 (Remarks to the Author):

The authors have adequately addressed the main points in my initial review. The only additional note I should make is that the fatty acid inhibition data in ref 25 newly added in this revision is questionable. It is perhaps unfair to point this out as this reference is peer reviewed, but I fear to cite it amplifies a message that is ambiguous.

We again are happy to have been able to address all the initial concerns of this reviewer. We appreciate the comment concerning reference 25: whilst this work has been peer reviewed/published and as such it would be inappropriate to remove this reference from the manuscript, we have adjusted the section of our work where this has been cited so as not to re-iterate the findings of this paper.

REVIEWERS' COMMENTS:

Reviewer #2 (Remarks to the Author):

My responses to the authors' second round of comments are marked with asterisks (*) within all the comments below:

Reviewer #2 (Remarks to the Author):

I concur with the authors that the resubmitted manuscript is a serious improvement from the original submission. The inclusion of the added experiments, specifically the knockouts, fluorescence tagging, ITC, MS/MS analysis, and more extensive in vitro assays has significantly strengthened the paper. In addition, the rewriting of the manuscript – and the appropriately tempered tone of their interpretation of the results – is to be commended. We again are very pleased to have been able to improve the manuscript in line with the comments of this reviewer.

I do have several comments/suggestions for the added experiments:

1) Figure 5 shows both the improved native-PAGE experiments with fluorescently tagged X and PCP-X domains as well as the ITC experiments with the X domain. The authors state in the main text: "These assays showed evidence of an interaction between the kistamicin Oxy enzymes and the X-domain independent of the presence of the adjacent PCP domain. This was evidenced by the formation of bands of higher molecular weight in the lanes where Xdomain and Oxy enzymes were present (Figure 5A-C)." However, from my interpretation, the data in Figure 5A-C clearly show substantially stronger OxyA/C associations with the PCPX domain than the X-domain alone. The authors, at the very least, should directly address this in the main text. And based on this fact, I would expect to see ITC data for OxyA/C associations with the PCP-X domain so readers can appropriately compare the sets of experiments.

We are happy to see that the reviewer agrees that the native-PAGE experiments clearly support interactions between the Oxy enzymes and constructs containing the X-domain. Our AUC data also supports that there is an interaction between the Oxy enzymes and the Xdomain. Beyond this, we have not attempted to quantify the native-PAGE data as we believe such interpretation would be prone to significant error. The purpose of including these experiments in the manuscript was to obtain evidence supporting the relevance of the OxyA/X-domain complex structure reported in this paper. It has clearly done this and further AUC experiments to determine PCP-X/ Oxy interactions would rather be probing the quantitative effect of the PCP on this interaction, which is not central to the discussion and significant quantity of results already contained in this manuscript.

*Quantitation of the native-PAGE analysis was never suggested. As the authors decided (and I think correctly so) to include this data as a main text figure, I would expect a reasonable qualitative analysis of the results. The justifiable interpretation that the PCP-X di-domain construct appears to have significantly stronger association with the Oxy domains from these experiments is not in disagreement with their structural data or the proposition that the X-domain is making primary contacts with the Oxy domains. The authors have not addressed a full qualitative assessment of these experiments in their response to my comments or in the main text. I still find that this should be addressed in the main text, and the editor can decide whether this is to the benefit of the readers.

2) The figure legends for Figures S17, S20, S22, and S24 need to more explicitly describe the respective figures. For instance, the authors should point out the respective loss or gain of peaks in the upper-right panels compared to the upper left panels (and mention the differently scaled y-axes) and what those results mean. Also, the X-axes are not all uniform in size as they should be. For the bottom left panels, the authors should describe what the 'Expected' column is denoting – it appears to be a theoretical isotopic distribution, but why/how that relates to the intensity is not clear.

We have altered these figure legends in line with the comments of this reviewer.

*This statement is incorrect. The figure legends beneath the figures have not been updated in any way.

Furthermore, we have also included a section of text (SI Page 23) by way of introduction to these data in the supplement to clarify and explain the data that is contained in each of these figures.

*This additional page does clarify much of what was asked to be put in the figure legends. I'm at a loss as to why this information is not added to the figure legends and placed here as it makes it less accessible and more confusing to the reader...the editor can comment on this if they desire.

3) For Figures S21 and S23, the chemical structures for the fragmented peptides with the various fragmentations should be added to these figures for clarity. Also, the authors should note in the figure legend of S23 the 'x5' and 'x50' sections of the spectrum. If these peaks are magnified in these mass ranges as I suspect, those mass ranges should be in in-set windows as to not look as if the spectrum is continuous.

We have included key ion structures for figures S21 and S23 as requested, with the remaining fragments common ions in mass spectrometry. Thus, we have not annotated every peak beyond including the names of each fragment, and have added two references explaining this nomenclature in the new MS introduction section now included in the supplement. We have also included a new version of figure S23 with the magnification removed and insets highlighting the key ions.

*While I still think these figures could be made more intuitive for the readers, I have no further comment on this data.

4) The authors should include MS/MS data for the bicyclic hexapeptide in their delta-kisN mutant. The ion count as seen in Figure S25 should be sufficient for fragmentation, and fragmentation between residues 3 and 4 of the hexapeptide would provide further evidence for the cyclization pattern and structure drawn in Figure 3.

We have included this data in the supplement as requested. In addition (and to avoid potential confusion with the interpretation of Figure 3), we now indicate that several bicyclic peptides have been identified from in vivo experiments (which was previously only mentioned in the text) and show the second ring formed as dotted/dashed lines to indicate this.

*Looks good.

5) Lastly, Figure S31 and Table S5 describe the size-exclusion chromatography experiments with the different interacting domains (with the MBP tags now cleaved off). From what I can tell from these graphs, these experiments show essentially no discernable/considerable interaction or appropriately sized complexes and should be explicitly pointed out in the main text. I am at a loss how this data is to be interpreted, especially without the relative sizes of the complexes noted based on a standard curve of molecular weight standards.

We agree that these data are not easily interpreted without considerable further effort, although they in no way conflict with our interaction hypothesis. In addition, the limited information content of these experiments is now essentially redundant when compared to that provided by the new labelled native-PAGE and Oxy/X AUC data now included in the manuscript (Figure 5) in response to referee comments. Given this, we have simply removed the size-exclusion experiments from the supplementary data. With significant new data incorporated into the supplement, we also have removed theazole binding figures previously included in the supplement as these are outside of the scope of this revised manuscript.

*I agree that removal of this data is best.

Response to Reviewers' comments:

We would once again like to thank the editorial staff and reviewer 2 for their time and analysis of our revised manuscript. We have been able to address the final list of concerns from this reviewer as noted below.

Reviewer #2 (Remarks to the Author):

1) ...regarding figure 5...

**Quantitation of the native-PAGE analysis was never suggested. As the authors decided (and I think correctly so) to include this data as a main text figure, I would expect a reasonable qualitative analysis of the results. The justifiable interpretation that the PCP-X di-domain construct appears to have significantly stronger association with the Oxy domains from these experiments is not in disagreement with their structural data or the proposition that the X-domain is making primary contacts with the Oxy domains. The authors have not addressed a full qualitative assessment of these experiments in their response to my comments or in the main text. I still find that this should be addressed in the main text, and the editor can decide whether this is to the benefit of the readers.*

We have added a sentence to the main text that notes the PCP domain may contribute to a stronger association to the Oxy enzymes based on these comments.

2) ...regarding figure legends for Figures S17, S20, S22, and S24...

**This statement is incorrect. The figure legends beneath the figures have not been updated in any way.*

**This additional page does clarify much of what was asked to be put in the figure legends. I'm at a loss as to why this information is not added to the figure legends and placed here as it makes it less accessible and more confusing to the reader...the editor can comment on this if they desire.*

Data has been removed from the additional page and is now added to each of the legends as requested

3) ... regarding Figures S21 and S23...

**While I still think these figures could be made more intuitive for the readers, I have no further comment on this data.*

We have tried to make the figures as intuitive as possible, however this is challenging due to the nature of the data

4) ...regarding MS/MS data for the bicyclic hexapeptide in their delta-kisN mutant...

**Looks good.*

We are pleased to have been able to include this data as requested

5) ...regarding size-exclusion chromatography experiments...

**I agree that removal of this data is best.*

We are pleased that the reviewer agrees with our approach of removing these experiments.